# Unconventional exciton evolution from the pseudogap to superconducting phases in cuprates

A. Singh [1], H. Y. Huang [1], J. D. Xie[2], J. Okamoto [1], C. T. Chen[1], T. Watanabe [3], A. Fujimori [1,4,5] ✉, M. Imada[6,7] ✉ & D. J. Huang [1,2,8] ✉

Electron quasiparticles play a crucial role in simplifying the description of many-body physics in solids with surprising success. Conventional Landau's Fermi-liquid and quasiparticle theories for high-temperature superconducting cuprates have, however, received skepticism from various angles. A path-breaking framework of electron fractionalization has been established to replace the Fermi-liquid theory for systems that show the fractional quantum Hall effect and the Mott insulating phenomena; whether it captures the essential physics of the pseudogap and superconducting phases of cuprates is still an open issue. Here, we show that excitonic excitation of optimally doped $Bi_2Sr_2CaCu_2O_{8+\delta}$ with energy far above the superconducting-gap energy scale, about 1 eV or even higher, is unusually enhanced by the onset of super-conductivity. Our finding proves the involvement of such high-energy excitons in superconductivity. Therefore, the observed enhancement in the spectral weight of excitons imposes a crucial constraint on theories for the pseudogap and superconducting mechanisms. A simple two-component fermion model which embodies electron fractionalization in the pseudogap state provides a possible mechanism of this enhancement, pointing toward a novel route for understanding the electronic structure of superconducting cuprates.

In cuprate superconductors above the superconducting transition temperature ($T_c$), various physical quantities show an enigmatic electronic excitation gap called a pseudogap[1,2]. In particular, the formation of the pseudogap in the antinodal region of the Brillouin zone manifests itself in the single-particle spectra measured with angle-resolved photoemission spectroscopy (ARPES)[3–9]. The mechanisms of the pseudogap formation and the superconductivity itself are core issues of high-$T_c$ superconductivity unresolved for decades[1,2,10–16].

To unravel the origin of the pseudogap, likely originating from electron correlation, it is crucial to clarify the two-particle dynamics in a momentum-energy resolved manner. Recent progress in resonant inelastic x-ray scattering (RIXS) spectroscopy has enabled an exploration of such desired particle-hole charge excitation to probe intrinsic exciton dynamics in the presence of the strong electron Coulomb interaction[17]. In the present work, we used RIXS to study electron-hole-pair (i.e., excitonic) excitation in cuprates and identified new phenomena that are critical to understanding the pseudogap and the superconductivity.

In order to understand the cuprates, one needs to take into account the fundamental fact that carrier-doped cuprates do not

[1]National Synchrotron Radiation Research Center, Hsinchu 30076, Taiwan. [2]Department of Electrophysics, National Yang Ming Chiao Tung University, Hsinchu 30093, Taiwan. [3]Graduate School of Science and Technology, Hirosaki University, Hirosaki, Aomori 036-8561, Japan. [4]Center for Quantum Science and Technology and Department of Physics, National Tsing Hua University, Hsinchu 30013, Taiwan. [5]Department of Physics, University of Tokyo, Bunkyo-ku, Tokyo 113-0033, Japan. [6]Research Institute for Science and Engineering, Waseda University, Shinjuku, Tokyo 169-8555, Japan. [7]Toyota Physical and Chemical Research Institute, Nagakute, Aichi 480-1192, Japan. [8]Department of Physics, National Tsing Hua University, Hsinchu 30013, Taiwan. ✉e-mail: fujimori@phys.s.u-tokyo.ac.jp; imada@g.ecc.u-tokyo.ac.jp; djhuang@nsrrc.org.tw

follow the conventional Fermi-liquid behavior. A possibility of breaking down the quasiparticle picture at the core of the Fermi-liquid theory has hence to be seriously pursued to understand the exciton dynamics. Among possible breakdowns, the scenario of electron fractionalization is one of the most impactful proposals, drawing attention to critical tests. The concept that an electron, being an elementary particle in vacuum, is splintered into two or more elementary "particles" in materials is already established in several phenomena in condensed-matter physics. Fascinating examples of electron fractionalization include: Laughlin's solution representing anyon excitation with fractionalized charge in the fractional quantum Hall effect[18,19], spin and charge solitons in poly-acetylene[20], and spin-charge separation in a one-dimensional Tomonaga-Luttinger liquid[21,22]. The splitting of an energy band into the upper and lower Hubbard bands, denoted by UHB and LHB, respectively, can be regarded as another typical example of electron fractionalization[16].

Electron fractionalization also embodies the idea of the duality of strongly correlated electrons. The duality is represented by, on one side, the conventional itinerant quasiparticles localized in momentum space connected with the overdoped Fermi-liquid, and by, on the other side, the electrons localized in real space leading eventually to the Mott insulator in the undoped limit. The duality shares a common concept of the coexistence of itinerant and localized electrons in a wider perspective proposed in the literature for metals, for instance in refs. [1, 23]. Regarding the localized nature of the electron, as is known in the Mott insulator, an electron is tightly bound to a hole requiring a charge gap to split into an electron and a hole. Such an electron bound to a hole is charge neutral in total and does not primarily interact with an electromagnetic wave. This binding may remain in the carrier-doped case.

A two-component fermion model (TCFM), which will be detailed below in this paper, was proposed to embody such a dual character of electrons[16,24,25]. One of the two constituents of the TCFM is the dark fermion, which is referred to as a "d fermion" and represents the localized side of an electron bound to a hole induced by doping. The dark fermion is expected to be more stable near the Mott insulator, while the other referred to as a "c fermion" representing the coherent metallic component of electrons becomes more stable in the overdoped region. The TCFM manifests this bistable nature of electrons in lightly carrier-doped systems. The idea of electron fractionalization and the TCFM has been studied to solve the mystery of the pseudogap phase in the high-$T_c$ superconducting cuprates[16,24,25] and is supported by the analysis of ARPES data[26]. Unlike other possible scenarios involving explicitly symmetry-broken phases with long-range order[10,13–15], the distinctive nature of the pseudogap mechanism based on electron fractionalization, as in the resonating valence bond scenario[11,12], does not require a spontaneous symmetry breaking, common to other types of fractionalization.

As the information from ARPES is limited to the dynamics of one particle, the analysis combined with two-particle spectroscopy is desired. In fact, the proposal of electron fractionalization supported by the ARPES measurement motivated a test by the RIXS measurement, because the enhancement of the RIXS intensity in the superconducting phase relative to the pseudogap phase was proposed uniquely when the electron fractionalization takes place[27].

To test the fractionalization idea, we performed Cu $L_3$-edge and O $K$-edge RIXS measurements on cuprates, and found an anomalous RIXS enhancement originating from the exciton dynamics in the superconducting phase relative to the pseudogap phase. Such an enhancement suggests a dramatic restructuring of the electronic structure likely associated with the strong electron correlation. For instance, a possible explanation is provided by the above-mentioned fractionalization[27]. Although the present measurement was motivated by the idea of fractionalization and qualitatively accounts for the enhancement, quantitative comparisons and stringent tests against

other possible interpretations are left for future studies. Aside from a unique identification of the origin, our experimental results of the RIXS intensity change establish the atypical exciton dynamics by itself, signaling the emergence of an unconventional electronic structure. The finding imposes a constraint on theories for the pseudogap and superconducting mechanism.

## Results

### Excitons revealed by Cu $L_3$-edge RIXS

To understand the present RIXS measurement, we discuss the underlying electronic structure calculated with the Hubbard model, which is expected to capture the essential high-energy spectral feature of the cuprates correctly. Figure 1a plots the single-particle spectral weight and dispersion of the pseudogap phase calculated with a doped square-lattice Hubbard model[28]. The doping of holes into a parent compound of cuprate creates a new low-energy state called in-gap band[29] separated by the pseudogap from the coherent part of the LHB that crosses the Fermi level $E_F$ in and near the nodal region, as one can see in the blowup shown in supplementary Fig. S1b[30]. The more localized UHB and the incoherent part of the LHB are located at ~4 and −2 eV, respectively.

RIXS probes collective excitation as well as excitonic excitations[27,31–34]. The latter can arise from intra- or inter-band transitions; their excitation depends on the incident photon energy. For hole-doped cuprates, the low-energy excitonic excitation results from transitions within the coherent LHB or between the coherent LHB and the IGB. Figure 1b plots Cu $L_3$-edge RIXS spectra of optimally doped $Bi_{2.1}Sr_{1.9}CaCu_2O_{8+\delta}$ (OP Bi2212) at varied incident photon energies. For the incident energy set to the maximum intensity of x-ray absorption spectrum (XAS), i.e., at $L_3$, we observed distinct Raman-like RIXS features centered about 0.5 and 2 eV. The RIXS feature about 2 eV is predominantly composed of $dd$ excitation formed by the transition of a Cu $2p$ core electron to the UHB and then the decay of another $3d$ electron to fill the $2p$ core hole. The spectral lineshape of $dd$ excitation reflects the local electronic structure of the Cu $3d$ states; it contains several components derived from transitions determined by the crystal field $10Dq$ and the tetragonal splitting of the $e_g$ and $t_{2g}$ orbitals.

The RIXS feature of energy loss below 0.5 eV is dominated by magnetic excitation, which becomes most pronounced when the incident x-ray energy is at the $L_3$ absorption peak[33–39]. As the x-ray energy is increased above $L_3$ + 0.6 eV, this RIXS feature appears to shift toward high-energy loss and evolves into a fluorescence-like excitation. The observed fluorescence-like shift is due primarily to the continuum of particle-hole excitation in the charge channel[33,34,39], i.e., an excitonic excitation created by the transition of an electron in the Cu $2p$ core level to the IGB (Fig. 1a, left panel), followed by the transition of an electron in the coherent LHB to the Cu $2p$ core level (Fig. 1a, right panel). The assignment of the peak structure in this energy range to the charge excitation (namely exciton formation) is corroborated by the fact that O $K$-edge spectra shown later also exhibit a similar corresponding peak, indicating that the observed structure arises from the charge channel as the single-spin-flip process is forbidden in the O $K$-edge RIXS.

For a typical in-plane momentum transfer $\mathbf{Q}_{\parallel} = (\pi, 0)$, the exciton is formed by an electron with energy $\omega_0$ above $E_F$ near the top of the IGB at $(\pi, \pi)$ and a hole below $E_F$ in the coherent LHB near $(0, \pi)$. Depending on the hole energy in the coherent LHB, the momentum transfer $\mathbf{Q}_{\parallel}$ does not necessarily bridge precisely between $(\pi, \pi)$ and $(\pi, 0)$, but some intermediate combination can occur in the entire momentum integration over the Brillouin zone. The photon energy $\omega$ determines the energy $\omega_0$ of the excited electron above $E_F$, i.e., $\omega = E_F + \omega_0$. The RIXS energy loss of this exciton is thus pinned by incident x-ray energy $\omega$, because the hole created in the coherent LHB with an energy close to $E_F$ has a dominant contribution from the region

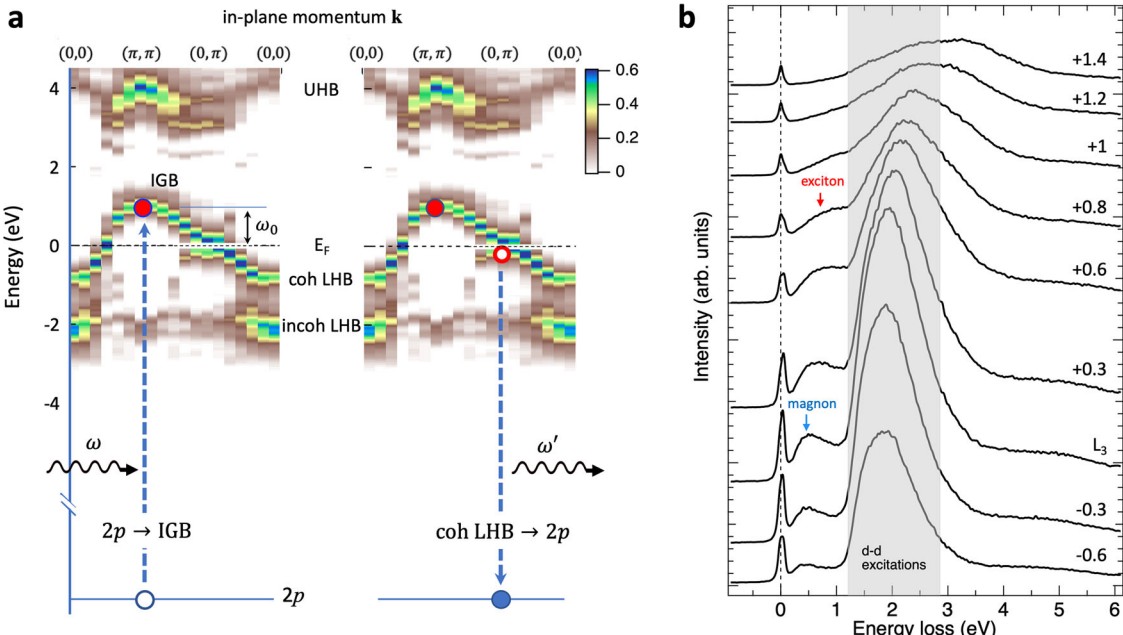

**Fig. 1 | Incident energy-dependent Cu $L_3$-edge RIXS of hole-doped Bi2212.**
**a** Illustration of excitonic excitation induced in RIXS. Left and right panels illustrate electronic transitions of resonant absorption and emission, respectively. Color intensity maps show single-particle spectral functions along a symmetric momentum line plotted by using data reproduced from Fig. 7a of ref. 28 at hole doping 12.5%. The hopping integral $t$ and the on-site Coulomb interaction $U$ used to simulate the cuprates are $t = 0.5$ eV and $U = 4$ eV, respectively. The in-plane momenta **k** are given in units of $1/a$, in which $a$ is the lattice parameter. The coherent part of the lower Hubbard band (LHB) and the incoherent part of the LHB are denoted by "coh LHB" and "incoh LHB," respectively. The left panel shows that a Cu $2p$ electron is excited to the in-gap band (IGB) near $(\pi, \pi)$ with an energy $\omega_0$

above $E_F$. The right panel illustrates that another electron in the coherent LHB decays to fill the $2p$ core hole by emitting another photon of energy $\omega'$. An exciton is then formed in the combination of red filled circle and red open circle. **b** Cu $L_3$-edge RIXS spectra of $\mathbf{Q}_\parallel = (\pi, 0)$ for various incident energies across the $L_3$ peak of the XAS. Spectra are plotted with a vertical offset for clarity. The region in gray indicates the energy loss region of local Cu $dd$ excitation. To identify the exciton peak, we have assumed the following: The energy loss in **b** has one-to-one correspondence to the energy difference between the red-filled and open circles in **a** when the momentum difference of the filled and open circles is $(\pi, 0)$. The incident energy in **b** is equal to the red-filled circle energy measured from $E_F$ in **a**.

of the largest density of states near $(0, \pi)$; the energy loss increases with increasing $\omega$.

For a quantitative analysis of the exciton energy, information about the energy $\omega_0$ of an excited electron is required. The main feature in the Cu $L_3$-edge XAS of hole-doped cuprates originates from the transition into the UHB. In the RIXS intermediate state, this UHB is pulled down by the $2p$ core-hole potential[40], as illustrated in Fig. S3a. Similarly, an excited electron in the IGB receives an attractive potential from the core hole to a somewhat smaller extent because of the character of spatially more extended than UHB. This effect leads us to estimate the energy position of $E_F$ in the Cu $L_3$-edge absorption spectrum through the fluorescence threshold from an extrapolation; we found that the Fermi level position corresponds to the absorption energy $L_3$. See supplementary Fig. S3.

**Suppression of excitons in the pseudogap phase**
We proceed to compare the change in the spectral feature of the exciton in RIXS between the superconducting and pseudogap phases. If the temperature is decreased below $T_c$, a superconducting gap $2\Delta \sim 80$ meV opens in OP Bi2212[41]. The excitonic excitation observed in our RIXS measurements has an energy scale much larger than the superconducting gap. Figure 2a displays Cu $L_3$-edge RIXS spectra in a wide range of energy loss with various incident photon energies for $\mathbf{Q}_\parallel = (\pi, 0)$. Interestingly, Fig. 2b shows that the exciton feature of energy loss $\Delta E \sim 0.8$ eV is enhanced in the superconducting phase, particularly for $\omega \sim 1$ eV above $L_3$, which induces a transition from $(0, \pi)$ to $(\pi, \pi)$. We attribute the difference 0.2 eV between $\Delta E$ and $\omega_0$ to the exciton binding energy in the final state. A similar enhancement of exciton of $\mathbf{Q}_\parallel = (\pi/2, 0)$ was also observed, but at a smaller $\omega_0$ (~0.8 eV above $L_3$) and with a smaller $\Delta E$ (~0.6 eV), as plotted in supplementary

Fig. S5. The variation in the intensity of elastic scattering, particularly for $\mathbf{Q}_\parallel = (\pi/2, 0)$, is due to the change in the tail of the scattering intensity from charge density waves, which is beyond the scope of the present paper[42,43].

To gain further insight into the origin of the enhancement in the exciton spectral weight, we resort to measuring RIXS of an overdoped (OD) sample with $T_c = 65$ K, which has no clear signature of the pseudogap. Figure 3 displays Cu $L_3$-edge RIXS of OD $Bi_{1.6}Pb_{0.4}Sr_2CaCu_2O_{8+\delta}$ (Pb-Bi2212) single crystal at temperatures above and below $T_c$. The shift of exciton energy from the OP to OD samples is about 0.5 eV, consistent with the expected chemical potential shift[44,45]. We observed no enhancement induced by the superconductivity in the exciton spectral weight of the OD Pb-Bi2212 sample. This leads us to suggest that the observed exciton enhancement in the superconducting state (or conversely the exciton suppression in the pseudogap state) in OP Bi2212 is connected to the existence of the pseudogap phase.

Figure 4 compares the RIXS intensity between the OP and OD samples by examining detailed temperature dependence across the superconducting transition for the RIXS feature arising from the exciton. The temperature evolution plotted in Fig. 4c shows a marked contrast between the OP and OD samples. In the OD sample, the intensity shows essentially no temperature dependence. In contrast, the OP sample shows first a decrease from 250 K to 150 K, where the pseudogap develops. Then further decrease of temperature below the superconducting fluctuation temperature $T_{scf} \sim 110$ K[46] shows a sharp increase of the intensity to the saturated value in the superconducting phase below $T_c$.

Excitons probed with the O $K$ edge consistently reveal the signature of unconventional restructuring of electrons. Figure 5a, b display RIXS spectra of OP Bi2212 for $\mathbf{Q}_\parallel = (\pi/2, 0)$ and show the same

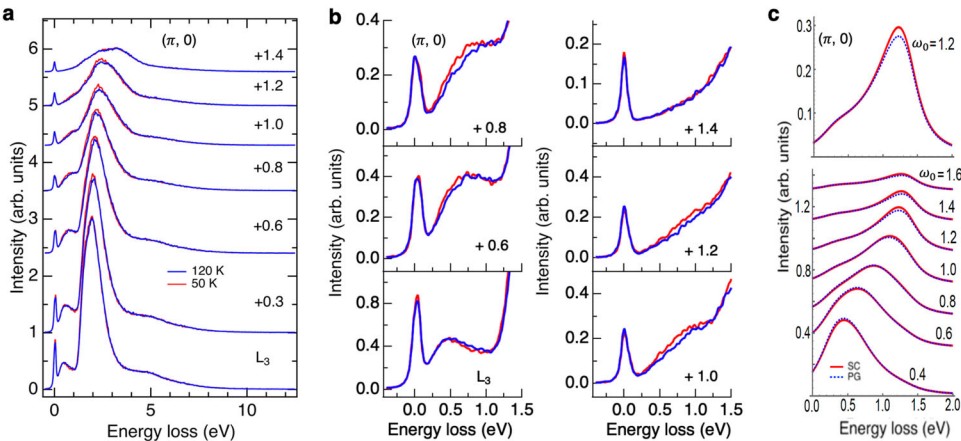

**Fig. 2 | Excitonic excitation of optimally doped Bi2212. a, b** RIXS spectra at selected incident photon energies for $\mathbf{Q}_\| = (\pi, 0)$ and temperatures above and below $T_c = 89$ K. The incident photon energy is denoted as its energy relative to the $L_3$ absorption peak in units of eV. Spectra above and below $T_c$ are normalized for energy loss integrated between 1.7 and 13 eV to highlight the difference caused by the change in temperature; the self-absorption effect of either the incident or the scattered photons was not corrected, because the change in temperature is not affected by the self-absorption. From the estimate shown in supplementary Fig. S3, $L_3$ corresponds approximately to $E_F$. **c** Calculated RIXS resulting from excitonic excitation in the superconducting (SC) (red curves) and pseudogap (PG) phases

(blue broken curves) for $\mathbf{Q}_\| = (\pi, 0)$ by using equations (1) and (4) as is detailed in Methods and in Supplementary information. The lower panel shows the spectra of selected incident energies $\omega_0$ and the upper panel is the zoom-in spectrum for $\omega_0 = 1.2$ eV to highlight the enhancement in the superconducting state clearly. The incident energy $\omega_0$ measured from $E_F$ is given in units of eV. The core-hole lifetime width $\Gamma$ and the broadening factor $\eta$ were set to 0.3 eV and 0.1 eV, respectively. For clarity, spectra in **a** are plotted with a vertical offset, of which are identical for each incident photon energy. Spectra in **c** are also presented with a vertical offset in a scheme similar to that of **a**.

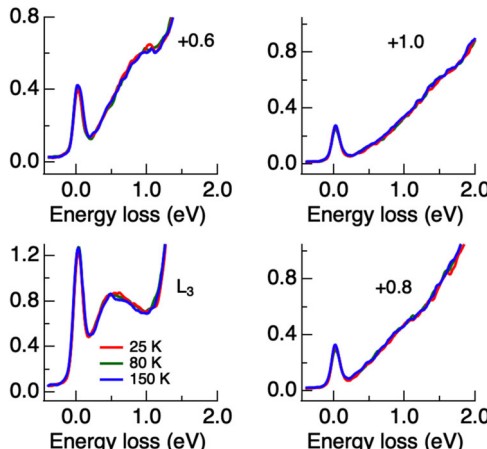

**Fig. 3 | Excitonic excitation in Cu $L_3$-edge RIXS of overdoped Pb-Bi2212.** RIXS spectra at selected incident photon energies for $\mathbf{Q}_\| = (\pi, 0)$ and temperatures above and below $T_c = 65$ K. The Pb-Bi2212 single crystal of $Bi_{1.6}Pb_{0.4}Sr_2CaCu_2O_{8+\delta}$ was with a hole doping level of $p = 0.22 \pm 0.005$. The incident photon energy is denoted as its energy relative to the $L_3$ absorption in units of eV. Spectra above and below $T_c$ are normalized for energy loss from 2 to 13 eV. The self-absorption effect of either the incident or the scattered photons was not corrected. See Fig. S7 for full-range RIXS spectra.

enhancement across the superconducting transition. In these RIXS spectra, the incident x-ray energy is expressed as the energy of the transition of the 1s core electron to the mobile hole band, the so-called Zhang-Rice singlet band (ZRSB). $A$ denotes the energy level corresponding to an absorption energy of 528.5 eV. Note that the ZRSB defined in the three-band Hubbard model is mapped onto the IGB plus coherent LHB in the framework of the single-band Hubbard model[47]. By analogy with the Cu $L_3$-edge RIXS, we assumed that the position of the bulk Fermi level $E_F$ corresponds to $A$ in the O $K$-edge XAS. The exciton enhancement in the O $K$-edge RIXS of $\mathbf{Q}_\| = (\pi/2, 0)$ occurs at energy loss 1.3 eV and also between 2.5 and 3 eV with incident x-rays of energy $A + 1$ eV. The 1.3-eV exciton feature is formed by an electron

excited to 1 eV above $E_F$ and a hole below $E_F$ created by the decay of an electron in the ZRSB near $(0, \pi)$ to the 1s core hole.

In conventional understanding, the temperature dependence from 50 to 120 K of the order of 5–10% in the weight of the RIXS intensity at the 1 eV exciton peak is surprising. In fact, all of the reports in the literature on the exciton measurements, either by RIXS[48,49] or optical conductivity[50,51], have proven that the temperature dependence is at most ~1% of the total spectral weight of the exciton peak. They emphasize either the peak suppression or shift through $T_c$ by a detailed analysis, but the relative change through $T_c$ is tiny (~<1%) and it is almost hidden in the background change of the 1% order or less. This proves that both the background temperature dependence and meaningful change below $T_c$ are very small.

We have observed unusual enhancement in the superconducting phase of OP Bi2212, although the temperature change and the exciton excitation energy differ by two orders of magnitude. In addition, our RIXS data at the Cu $L_3$- and O $K$-edges show consistent temperature dependence. This is by itself a surprising result, and must reflect a dramatic change of the electronic structure in that temperature range due to a specific mechanism and cannot be a simple background effect. Note that the enhancement observed both in Cu $L_3$ and O $K$-edges evidences that are ascribed to the charge excitation and not to the single-spin flip magnetic excitation.

**Two-component fermion model analysis**

The enhancement of excitons at energies much higher than the superconducting gap is highly nontrivial. To the best of our knowledge, the atypical change in the spectral feature described above cannot be explained with a single-component model as plotted in supplementary Fig. S6. In this simple single-component case, in proceeding from the normal state to the superconducting state with the BCS mean-field order, the total weight of the quasiparticle component remains unchanged; the change in the high-energy region should be negligible, although the single-particle spectral weight near $E_F$ is decreased because of the superconducting gap formation. We shall also refer to previous studies that most of the exciton measurements in the literature show essentially no temperature dependence compared to the present conspicuous enhancement below $T_c$. This has

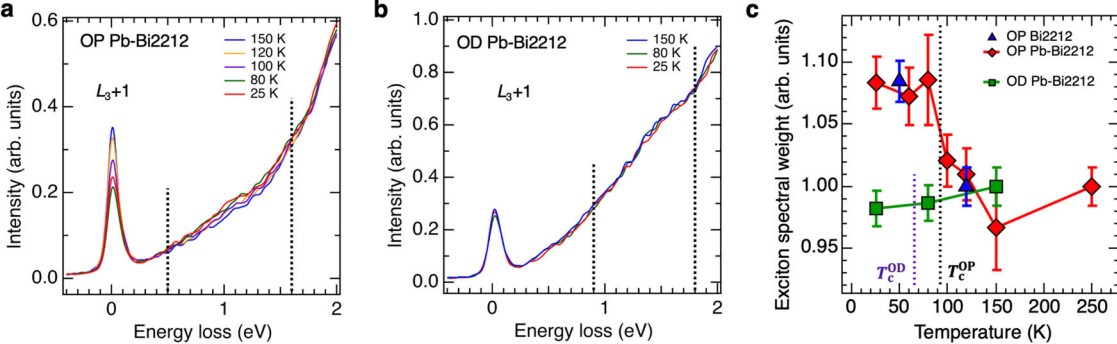

**Fig. 4 | Evolution of the exciton spectral weight across the superconducting transition. a, b** Cu $L_3$-edge RIXS induced by incident photons of energy tuned to 1 eV above the $L_3$ absorption energy with $\mathbf{Q}_\parallel = (\pi, 0)$ for OP and OD Pb-Bi2212, respectively. The normalization scheme of spectra is the same as those described in Figs. 2 and 3. The dotted lines indicate the energy range selected for spectral-weight integration. **c** Evolution of the exciton spectral weight defined as the integration of RIXS between 0.5 and 1.6 eV for OP and between 0.9 and 1.8 eV for OD samples, respectively, as indicated by dotted lines in **a, b**. The exciton spectral weight is normalized to one for 250 K and the error bars are estimated from the variations in exciton spectral weight deduced by using different energy loss regions. See Supplementary Information for the analysis details of the error bars. The exciton spectral weights of the OP pristine Bi2212 at two temperatures are also included in **c**, as plotted in blue triangles. The vertical dotted lines indicate the transition temperatures $T_c$ of OP and OD Pb-Bi2212 crystals.

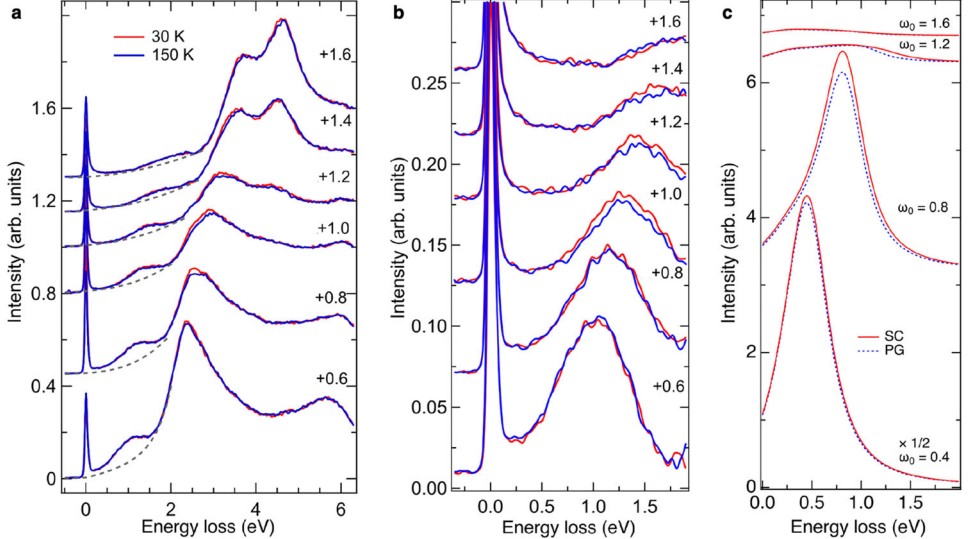

**Fig. 5 | Exciton excitation in O $K$-edge RIXS of OP Bi2212. a** O $K$-edge RIXS spectra at selected incident photon energies for $\mathbf{Q}_\parallel = (\pi/2, 0)$ and temperatures above and below $T_c$. The incident photon energy is denoted as its energy relative to the absorption energy $A$ in units of eV. The tail of the RIXS feature created by the transition from the incoherent LHB to the IGB is depicted by a dashed line. **b** Zoom-in spectra of **a** after removing the tail of RIXS induced by the transition from the incoherent LHB to the IGB. **c** Calculated RIXS for $\mathbf{Q}_\parallel = (\pi/2, 0)$ with the same condition as in Fig. 2, except for $\Gamma = 0.07$ eV and $\eta = 0.1$ eV. For clarity, all spectra are plotted with a vertical offset, of which are identical for each incident photon energy in the same plot.

motivated analyses by the two-component fermion model based on the electron fractionalization because the enhancement was predicted in the TCFM analysis[27].

The electron bistability sketched in the introduction induces two different tendencies. One is the phase separation into microscopically carrier rich (dominated by $c$ fermion) and poor regions (dominated by $d$ fermion) or inhomogeneity in real space such as the stripe order as is widely observed in the cuprates. The other is a resonant combination of $c$ and $d$ fermions stabilized uniformly in real space as is represented by the TCFM formalism. We focus on the latter. More explicit description of TCFM (Eq. (1)) and its consequence for the RIXS spectra is found in Methods (See also Supplementary Information for the physical properties of TCFM).

In this non-interacting model, the hybridized (resonating) $c$ and $d$ fermions generate the bonding and antibonding bands, which correspond to the coherent LHB and the IGB, respectively. This simple framework well describes the essence of electron fractionalization[16], which enables us to understand the mechanism of the pseudogap of cuprates[30]. The hybridization occurs in such a way that the fractionalization and resultant pseudogap are most prominent around the antinodal points, i.e., $(\pi, 0)$ and $(0, \pi)$. Although the pseudogap is ascribed to some complex many-body effect in the framework of the one-component interacting fermion system, in the TCFM it is interpreted as a simple one-body hybridization gap of the two constituents, and no spontaneous symmetry breaking is required.

The TCFM model qualitatively captures the essence of the change in the exciton spectral weight despite the simplification of calculations for RIXS spectra. Below we focus on the inter-band transitions which lead to an exciton formed between an IGB particle and a coherent LHB hole, and compare the exciton spectra between the experimental data and the TCFM calculations. Figures 2c and 5c plot, respectively, calculated Cu $L_3$-edge and O $K$-edge RIXS spectra for $\mathbf{Q}_\parallel = (\pi, 0)$ & $(\pi/2, 0)$

in the pseudogap and superconducting phases using the TCFM. The calculation reproduces the increase of the energy loss with $\omega_0$ in agreement with the RIXS results. The calculations also explain the enhancement of the RIXS intensity at around $\omega_0 \sim 1\,eV$ in the superconducting phase.

The mechanism of the enhancement shown by the TCFM calculation is conceptually explained as follows[27]. The electronic states in the filled part near the antinodal region in the pseudogap phase contain a large weight of the dark fermion component. However, since the transition to the core hole in the RIXS process is allowed only for the quasiparticle (quasihole) component and not for the dark fermion, suppression of the RIXS intensity as compared to that above the pseudogap temperature occurs. As the superconductivity is realized by the Cooper pairing of conventional electrons, the fractionalization is suppressed, resulting in the recovery of the quasiparticle component. It leads to an enhanced transition probability to the core hole; in turn, the RIXS intensity increases.

## Discussion

Although the spectral profiles of experimental and calculated Cu $L_3$-edge RIXS around 1 eV plotted in Fig. 2 look different at first glance, the difference can be attributed to the contributions from the strong tail of the $dd$ excitation peak at 2–3 eV, paramagnons, and other excitations. One can identify the exciton peak around 1 eV and compare its energy with the TCFM calculation. For O $K$-edge RIXS, its spectral background is simpler; one can extract the exciton peak through a background subtraction and compare its spectral profile with the TCFM calculation as shown in Fig. 5b, c. These comparisons led us to find an agreement between the experiment and calculation in the following sense: (a) The exciton peak energy in the RIXS spectra has an overall and good agreement between the measurement and calculation. For instance, at the Cu $L_3$ edge, the peak energy $\omega_{peak}$ is nearly equal to the incident energy $\omega_0$, namely, $\omega_{peak} \sim \omega_0$ in both experiment and calculation at the momentum transfer $(\pi, 0)$. The peak width is ~0.5–1.0 eV, which is also in the agreement between experiment and calculation. (b) The enhancement in the calculation at the peak energy is 7.6% for the Cu $L_3$ edge at $\omega_0 = 1.2\,eV$ and $\omega_{peak} = 1.23\,eV$. In the case of the experiment, as one sees in Fig. 4c, the enhancement at temperatures far below $T_c$ is ~7–10% relative to the pseudogap region ~110–150 K.

In the above comparisons, the parameters in the TCFM model are solely chosen to reproduce the ARPES data without further adjustment. Because ARPES data do not contain the unoccupied part, TCFM parameters may contain larger errors for those related to the unoccupied degrees of freedom. Nevertheless, we obtained the essential agreement of the exciton energy (a) and the enhancement rate (b) between the TCFM calculation and RIXS data. By considering the simplicity of TCFM without parameter adjustment, this agreement is highly nontrivial.

Although the TCFM accounts for the enhancement from the pseudogap phase to the superconducting phase, the measured enhancement is not isolated from other resonance peaks such as the $dd$ excitation, while such a complication is ignored in the TCFM. Therefore, to establish the origin of the enhancement firmly, further studies are needed. Interpretations not associated with fractionalization are equally called for to deepen the understanding of this enhancement. RIXS data contain richer information in the unoccupied part of the single-particle spectra as well as the two-particle exciton spectra. Present combined analysis assures that integrated analyses of ARPES and RIXS together with other spectroscopic data will provide us with a more comprehensive understanding of the electron correlation effect in general in the future.

As for the different exciton energies between the O $K$ edge (~1.3 eV) and the Cu $L_3$ edge (~0.8 eV) with $\omega_0 = 1\,eV$, one possible scenario is to take into account the character of the IGB in more detail. According to a TCFM proposal, the IGB could be a fermionic

component of the weakly bound Wannier exciton, in which the excited electron is loosely bound to a nearby hole[16]. The repulsive interaction between the O $1s$ core hole and the bound hole residing away from the excited electron can increase the exciton energy. In this interpretation, the IGB state can be regarded as a consequence of the screening of an electron in the unoccupied level by a hole and might bridge the picture of fluorescence in the actual physical process. For the Cu $L_3$ edge, the core hole works instead as an attractive potential for the Wannier exciton, because the Cu core hole dominantly interacts with not the bound hole but the excited electron at the same Cu site, and decreases the energy loss. This effect explains the energy difference of the exciton between Cu $L_3$ and O K-edges beyond the present simplified TCFM analysis.

We observed the same enhancement of an exciton excitation with an energy loss centered at about 2 eV, overlapping the tail of the Cu $dd$ excitation; see Fig. 2a. This enhancement arises from an effect of the Cu $2p$ core-hole potential, namely, a shake-up of the photo-excited electron from the IGB to the UHB. As shown in Fig. 1b, the onset of the electron-hole pair excitation shares the same resonant energy of incident photons with the $dd$ excitation, because the $2p$ core-hole potential pulls down the UHB in the intermediate state of RIXS. Therefore, hybridization between the IGB and the UHB is increased, inducing the shake-up from the IGB to the UHB. In other words, the electron is first excited to the IGB near $(\pi, \pi)$ from Cu $2p$ and then further excited to the UHB through the shake-up process. A high-energy exciton is hence formed by an electron in the UHB and a hole in the coherent LHB near $(0, \pi)$. Although the TCFM is a low-energy effective model to describe the IGB and the coherent LHB and takes no account of the high-energy exciton involving the UHB, the same mechanism ascribed to the electron fractionalization near the Fermi level discussed above is expected to account for the enhancement of the high-energy exciton in the same manner. The physics of electron fractionalization will equally apply to an exciton with an upper Hubbard band electron since the hole resides in the same coherent LHB. The extension of the TCFM to include the upper Hubbard band, however, requires a substantial extension of the model. The parameters needed for the extension are not available so far from the ARPES data. Therefore, this is not within the scope of the present work and is left for future studies. Again, it is open to other interpretations if possible.

In contrast, the spectral-weight transfer from high to low energies in the superconducting state relative to the normal phase was observed in the optical conductivity[50], though a small amount. Such transfer, also suggested by the uniform mixing of the UHB and LHB in a simple model[52], may imply a mechanism similar to the fractionalization here at work. However, suppose the UHB-LHB mixing uniformly enhances the LHB weight in the superconducting state; it is unclear why the enhancement is not observed at low energies around the elastic peak in the experimental RIXS spectra contrary to the naive expectation from this picture.

Interestingly, previous optical data[53] also show enhancement of the imaginary part of the dielectric constant in the order of 10% for an underdoped sample for the energy around 3–3.5 eV. This result is indeed consistent with ours in terms of the fractionalization mechanism and further supports the universal feature of electron fractionalization, because the same mechanism of the enhancement in the superconducting phase works as the enhancement for LHB-UHB excitons observed around 2–3 eV in Figs. 2 and 5, as is expected from the optical transition from the coherent LHB at $(0, \pi)$ to the UHB at the same momentum that has a substantial density of states (see Fig. 1a).

In conclusion, Cu $L_3$- and O K-edge RIXS spectra of OP Bi2212 consistently show an enhancement of the exciton intensity in the superconducting phase relative to the normal phase with a pseudogap. The exciton peak energy and the enhancement is more or less consistent with the prediction of the TCFM that the electron fractionalization is suppressed in the superconducting phase. The quasiparticle

component dominant above the pseudogap temperature $T^*$ partly recovers below $T_c$. We have discussed the enhancement without assuming any symmetry breaking because a simple mean-field-type symmetry breaking does not generate the dark fermion that is responsible for the enhancement. Therefore, spontaneous symmetry breaking such as charge order and nematicity proposed as mechanisms of the pseudogap would be difficult to explain the enhancement of RIXS intensity in the superconducting phase. In this sense, the present enhancement will pose severe constraints on the interpretation of the pseudogap as well.

The exciton peaks in the RIXS spectra show a substantial broadening. If intrinsic, it gives a rough estimate of the exciton lifetime in the order -0.1 eV$^{-1}$. The possible interpretation of the dark fermion coupled to the exciton mentioned above offers a consistent picture, and the dark fermion lifetime is estimated in the same order. The large self-energies of excitons and dark fermions make it necessary to take into account the interaction of the dark fermion in future more elaborate studies. This enables us to disclose the nature of the dark fermion and its strong quantum entanglement on the microscopic level beyond the TCFM and opens a possible route to studies on Planckian fluids[54–56] as suggested in the analysis of ARPES data[26]. This is an intriguing future issue.

Further studies are called for including extensions beyond the TCFM particularly to understand the interaction and damping of the fractionalized particles to test the fractionalization mechanism more quantitatively stringently. As we repeatedly emphasized above, a critical comparison with possible alternative mechanisms other than electron fractionalization is desired as well, if it exists.

## Methods

Single crystals of pristine Bi$_{2.1}$Sr$_{1.9}$CaCu$_2$O$_{8+\delta}$ (Bi2212) and Pb-doped Bi$_{1.6}$Pb$_{0.4}$Sr$_2$CaCu$_2$O$_{8+\delta}$ (Pb-Bi2212) were grown in the air using the traveling solvent floating zone method[57]. The hole doping levels of the crystals were obtained using Tallon's empirical relation[58]. The crystals were then annealed under oxygen partial pressures of 100 Pa at 600 °C and 2 Pa at 600 °C to make them OP with $T_c$ = 89 and 93 K for Bi2212 and Pb-Bi2212, respectively. The doping level of the OP crystals was $p$ = 0.16 ± 0.005. An overdoped Pb-Bi2212 crystal was also prepared with $T_c$ = 65 K and $p$ = 0.22 ± 0.005.

We conducted Cu $L_3$-edge and O $K$-edge RIXS measurements at the AGM-AGS spectrometer of beamline 41A at Taiwan Photon Source[59]. XAS was recorded at normal incidence with $\sigma$ polarization using the total-electron-yield method. RIXS spectra across $T_c$ were recorded with $\sigma$ polarized incident X-rays of which the polarization was perpendicular to the scattering plane, as illustrated in Fig. 6. The scattered x-rays were detected without a polarization analysis.

We calculated the exciton intensities in RIXS for the superconducting and pseudogap phases with $\mathbf{Q}_\parallel = (\pi, 0)$ and $(\pi/2, 0)$ using the TCFM defined by the following Hamiltonian:

$$H = \sum_{k,\sigma} \Big[ \epsilon_c(k) c^\dagger_{k,\sigma} c_{k,\sigma} + \epsilon_d(k) d^\dagger_{k,\sigma} d_{k,\sigma}$$
$$+ \Lambda(k) (c^\dagger_{k,\sigma} d_{k,\sigma} + \text{H.c.})$$
$$+ (\Delta_c(k) c^\dagger_{k,\sigma} c^\dagger_{-k,-\sigma} + \Delta_d(k) d^\dagger_{k,\sigma} d^\dagger_{-k,-\sigma} + \text{H.c}) \Big], \tag{1}$$

where the fermion $c$ represents the original quasiparticle with the dispersion $\epsilon_c(k)$ at the momentum $k$ in a form of a non-interacting Hamiltonian. The dark fermion represented by $d$ with the dispersion $\epsilon_d(k)$ emerging from the strong correlation of the electrons hybridizes to the fermion $c$ via the coupling $\Lambda(k)$. We employed a square lattice with simple dispersion for $\epsilon_c(k)$ and $\epsilon_d(k)$ as

$$\epsilon_c(k) = -(2t_{c1}(\cos k_x + \cos k_y) + 4t_{c2} \cos k_x \cos k_y) + \mu_c,$$
$$\epsilon_d(k) = -(2t_{d1}(\cos k_x + \cos k_y) + 4t_{d2} \cos k_x \cos k_y) + \mu_d, \tag{2}$$

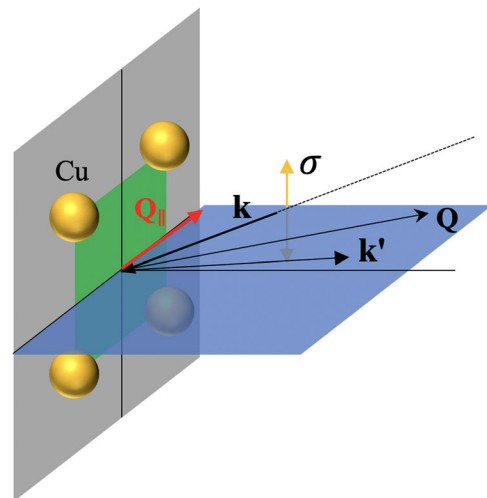

**Fig. 6 | Scattering geometry of RIXS measurements.** The scattering plane is defined by the incident and scattered wave vectors, $\mathbf{k}$ and $\mathbf{k}'$, respectively. The projection of the momentum transfer onto the CuO$_2$ plane, $\mathbf{Q}_\parallel$, is along the antinodal direction $(\pi, 0)$; $\mathbf{Q}$ is defined as $\mathbf{k}' - \mathbf{k}$.

where $t_{c1}$ ($t_{d1}$) and $t_{c2}$ ($t_{d2}$) represent the nearest-neighbor and next-nearest-neighbor hoppings of $c$ ($d$) fermion, respectively. The terms proportional to $\Delta_c(k)$ and $\Delta_d(k)$ represent the anomalous part in the mean-field approximation emerging in the superconducting state. Here we assume the simple $d$-wave superconducting gap

$$\Delta_c(k) = \frac{\Delta_{c0}}{2}(\cos k_x - \cos k_y),$$
$$\Delta_d(k) = \frac{\Delta_{d0}}{2}(\cos k_x - \cos k_y). \tag{3}$$

and the hybridization in the form $\Lambda(k) = \Lambda_0 + \Lambda_1 (\cos k_x + 1)(\cos k_y + 1)$. The parameter values were fitted to reproduce the ARPES[60] data together with its machine learning analysis[26] and STM data[61] as $t_{c1} = 0.1953$, $t_{c2} = -0.0762$, $t_{d1} = 0.0100$, $t_{d2} = -0.0036$, $\mu_c = 0.2875$, $\mu_d = 0.0105$, $\Delta_{c0} = 0.02$, $\Delta_{d0} = 0.067$, $\Lambda_0 = 0.0658$ and $\Lambda_1 = -0.014$ in the unit of eV. For the details of the TCFM calculations, see Supplementary Information and refs. 25, 27.

The RIXS intensity at the momentum $\mathbf{Q}$ and frequency $\omega$ was calculated using the formula

$$I_{\text{RIXS}}(\mathbf{Q}, \omega, \omega_0; \sigma, \rho) \propto \sum_l |B_{li}(\mathbf{Q}, \omega_0; \sigma, \rho)|^2 (E_l - \omega + i\eta)^{-1}, \tag{4}$$

$$B_{li}(\mathbf{Q}, \omega_0; \sigma, \rho) = \sum_{m,j} e^{i\mathbf{Q} \cdot \mathbf{R}_m} \chi_{\rho,\sigma} \langle l|c_{m\sigma}|j\rangle \langle j|(\omega_0 - E_j + i\Gamma)^{-1}|j\rangle \langle j|c^\dagger_{m\rho}|i(k=0)\rangle, \tag{5}$$

where $|i(k=0)\rangle$ is the ground state and $\chi_{\rho,\sigma}$ is the spin ($\rho$ and $\sigma$) dependent matrix representing the product of two dipole matrix elements for absorption and emission. The core-hole lifetime width and exciton lifetime width are denoted by $\Gamma$ and $\eta$, respectively. The x-ray excites a core electron to the level $\omega_0$ measured from $E_F$. Here, $c^\dagger_m$ is the local electron creation operator at the $m$-th site with the coordinate $\mathbf{R}_m$. The one-electron excited state represented by $|j\rangle$ is an eigenstate of the Hamiltonian (1) with energy $E_j$. All of them are relevant to the experimental setup of the crystal axes, x-ray polarization, and momentum transfer.

## Data availability

All data generated or analyzed during this study are included in this published article and its supplementary information files.

## Code availability

The custom code and mathematical algorithm used in this study are available from M.I. (imada@g.ecc.u-tokyo.ac.jp) upon request.

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

## Acknowledgements

This work was supported in part by the Ministry of Science and Technology of Taiwan under Grant no. 109-2112-M-213-010-MY3 and 109-2923-M-213-001. We also thank the support of KAKENHI Grant no. 15H02109, no. 16H06345, no. 19K03741, no. 20K03849, no. 22K03535, and no. 22H05114 from JSPS. One of the authors, T.W., was supported by a Hirosaki University Grant for Distinguished Researchers from fiscal year 2017 to fiscal year 2018. This research was also supported by MEXT as "Program for Promoting Researches on the Supercomputer Fugaku" (Basic Science for Emergence and Functionality in Quantum Matter - Innovative Strongly Correlated Electron Science by Integration of Fugaku and Frontier Experiments -, JPMXP1020200104) together with computational resources of supercomputer Fugaku provided by the RIKEN Center for Computational Science (Project ID: hp200132, hp210163, and hp220166).

## Author contributions

D.J.H. and M.I. coordinated the experimental and theoretical works, respectively. A.S., H.Y.H., J.D.X., J.O., D.J.H., and C.T.C. conducted the RIXS experiments. A.S., H.Y.H., J.D.X., D.J.H., and A.F. analyzed the data. M.I. performed the TCFM calculations. T.W. synthesized and characterized the Bi2212 samples. D.J.H., M.I., A.F., and A.S. wrote the manuscript with inputs from other authors.

## Competing interests

The authors declare no competing interests.
