## [Peer Review File · Nature Communications]

Unconventional exciton evolution from the pseudogap to superconducting phases in cupratesReviewers' comments:

Reviewer #1 (Remarks to the Author):

The authors of the manuscript by Singh et al. aim to study the exciton dynamics originating from electron fractionalization of a superconducting cuprate into the lower and upper Hubbard bands. Resonant Inelastic X-Ray Scattering (RIXS) experiments at Cu L3 and O K-edges on optimally hole doped $\text{Bi}_2\text{Sr}_2\text{CaCu}_2\text{O}_{8+\delta}$ (Bi-2212) display a medium-energy response around 1 eV framed as excitonic excitations much beyond the superconducting-gap energy scale. The main experimental finding of the study is that these excitations are unusually enhanced upon entering the superconducting state. The authors see this as a clear proof of the involvement of such excitons in the superconductive mechanism.

The experimental RIXS data presented in the paper are certainly of high quality and the observation of such a spectral weight enhancement across the superconducting transition is certainly interesting. The experimental test of a hypothesis made from a state-of-the-art model calculation is an original result. I find that the topic could in principle be suitable for high profile journals like Nature Communications. However, the paper largely describes an experimental test of the prediction of inelastic scattering enhancement made in already published theory papers (ref. 23 and 26) by a simple two-component fermion model, which is based on the electron fractionalization concept. By this only part of the paper is fully original. For instance, Fig. 1 is recycled from the previous publication. A further crucial shortcoming of the manuscript is that the physical picture suggested in ref. 23 and 26 is described in a way that is not appropriate for the general audience. It would be required for a publication in Nature Communications that a broad audience can follow the proposed physical concept without consulting several theory papers delivering the background. The referee finds the concept impossible to follow and comprehend without consulting these previous publications. Coming to the experimental results, the description of the incident energy dependence of the RIXS data in Fig. 2 is very difficult to follow as well. The development of the magnon excitations with a Raman response around the Cu L3 resonance into a broad exciton showing a fluorescence behavior is very bad explained. Apparently, this invokes an electron-hole pair mechanism that would need to be described in detail. For a broader audience a schematic illustration of such scattering mechanism could possibly illuminate such complicated processes. The authors seem to claim that the enhanced spectral response must be due to charge scattering. The argument for this lies in an ad hoc explanation that the changing spectral weight must be of charge origin, since the 0.5 eV magnon response does not show any spectral enhancement with temperature. This argument is inconclusive, as indeed the response around 0.5 eV energy loss is known to be composed of both spin and charge excitations, when no polarization selectivity is applied in the detected channel. Looking how well the experimental data agree with the calculations in Fig. 3 is also not easy, as these panels present a zoom into different energy scales. At best, there is a qualitative agreement. The authors claim an enhancement in the order of 5-10%, but do not show any quantitative analysis. Actually, the clearer temperature development is seen not in the exciton region around 1 eV, but rather in the spectral region of the dd-excitations above 2 eV, which seems to be much more important than the region around 1 eV. Unfortunately, the model does not describe this spectral region. Therefore, the claimed "stringent test of a theoretical prediction" is by no means unambiguous.

In summary, the paper reports interesting medium-energy scattering data by RIXS across the superconducting transition that are not conclusively analyzed with a previously developed theory model. The paper lacks the urgency required by Nature Communications and should better be submitted to a more specialized journal. This will require, however, a total rewrite of the paper that explains the processes and spectral development much more clearly.

Reviewer #2 (Remarks to the Author):

In the manuscript by Singh et al., the authors report on a resonant inelastic X-ray scattering (RIXS) experiment carried out on an optimally-doped Bi2212 single crystal. Their central finding is made when the incident photon energy is detuned to the higher-energy side of copper L3 and oxygen K absorption edges by about 1 eV, where their experimental configuration (sigma-polarization grazing incident) is known to allow for observations of charge excitations (which the authors call excitons) arising from the generation of particle-hole pairs.

The claimed novel observation is that the intensity of the signal, primarily of the fluorescence (rather than the 'Raman') type, is stronger at $T = 50$ K than at 120 K, the former of which is below the superconducting critical temperature T_c of the sample. It is argued that this below- T_c enhancement is caused by a competition between superconductivity and pseudogap physics, the latter of which is presumed to cause quasiparticle fractionalization, such that when superconductivity helps restore the quasiparticle component, the RIXS intensity is enhanced. It is concluded that the observed temperature-dependent signals at relatively large nominal momentum transfers have been experimentally missed so far, and that the phenomenon proves the involvement of high-energy excitons in superconductivity; this in turn supports the so-called two-component fermion model, in which the pseudogap phase is theoretically known to feature fractionalized quasiparticles.

The paper is well-written, with a clear motivation for the study and smooth logic flow. I also tend to agree that if superconductivity can indeed enhance a new type of resonant scattering (or fluorescence) intensity, at a much higher energy than the pairing gap and with peculiar characteristics allowing for pin-pointing the underlying mechanism to be within the two-component fermion model (or even just any models that ensures quasiparticle fractionalization), it would be a result novel enough to warrant publication in Nature Communications. Here, in agreement with the authors' own claim, I consider the peculiar characteristics to include (i) prominence of the effect beyond physically trivial interpretations, (ii) observable only at high momentum transfers, (iii) enhancement below T_c , and (iv) linkage to the pseudogap phase and its competition with the superconductivity. Unfortunately, I see a series of difficulties with essentially all of these aspects, and remain completely unconvinced of the authors' claim.

1) The authors use a normalization method to compare data sets below and above T_c , as explained in Fig. 3 captions: the normalization keeps intensities at larger than 5 eV energy loss constant. This procedure leads to a very suspicious consequence, in that the d-d excitation peak area changes considerably, under precisely the same detuned incident energies where the largest enhancement (below T_c) is seen for the 'excitons'. An alternative way of normalization, which is quite commonly employed in Cu L3 RIXS experiments, is to use the d-d area (keeping the d-d peak area constant below and above T_c). What would the authors get if they do this instead? I am particularly concerned by the fact that their displayed RIXS energy spectra are only up to ~ 6.5 eV, and the integrated spectral weight between 5 eV and 6.5 eV is only a small fraction of the total intensity between 0 and 6.5 eV. Then, a small normalization systematic error could result in a large bias on the important peaks' (both d-d and exciton) intensities. Moreover, it is well known that RIXS intensity needs to be corrected for sample's self-absorption (for the scattered photons), and the strength of such absorption can be strongly energy-loss-dependent too, especially if the incident energy is detuned to the high-E side. Since the authors' conclusion heavily depends on spectral comparison at a detailed level, and because the observation involves scattered photons that are on-resonance (detuned by ~ 1 eV, then with ~ 1 eV energy loss), these technical aspects along with the uncertainties they bring need to be handled with special care, but I do not find them being carefully discussed at all. For me, this severely hampers the credibility of the entire pack of the experimental results.

The authors made an argument related to shake-off effects to explain the change in the d-d peak with temperature, which makes no sense to me at all: With such shake-off, the d-d peak enhancement should not be restricted to only the 'correct' incident photon energies. To me it looks very much like a systematic error with the normalization procedure. Please check.

2) The authors studied only two momenta ($\pi,0$) and ($\pi/2,0$), and at both places they found an enhancement. To really show that the effect pertains to large Q transfer only, and to make a case consistent with previous studies reporting no such effects, I expect the authors to show a much weaker effect or absence of effects at smaller Q using their same sample, same experimental set up.

3) Data at only two temperatures are insufficient for claim a connection to T_c . If the authors really believes in such a connection, they should at least be able to show a full variable-T data set by focusing on just one Q and one good incident energy. Being able to show such a data set would also significantly help resolve my point 1) above, because when data at many temperatures show that only the exciton intensity is strongly changing near T_c , it would be unlikely to be due to systematic errors related to normalization.

4) I know RIXS is a demanding type of experiment, but if the claimed novel phenomenon does require a competition between superconductivity and pseudogap phase, studying an overdoped sample (and observing absence of similar effects) would be a very good bench-marking experiment. It is also well-known (see, e.g., Minola et al., PRL 119, 097001 (2017), including supplementary materials Fig. 2) that the charge excitations are more prominent on the overdoped side, so the study should at least be feasible.

In summary, I am rather unconvinced of the technical correctness (primarily experimentally) of the results presented in their present form. I believe that the above issues need to be overcome before the work can be published.

Reviewer #3 (Remarks to the Author):

The manuscript presents the comparison of RIXS spectra of Bi2212 measured across the Cu L3 and O K edges with theoretical spectra of the particle-hole channel calculated with the “two-component fermion model” (TCFM). Whereas the latter results were introduced recently by one of the authors in reference 26, although for a limited number of cases, the former are more original. The main observation is that RIXS spectra measured at 50K and 120K, ie below and above the superconducting transition temperature of the chosen sample, visibly differ in intensity in the 0.3-2.5 eV energy loss range when the incident photon energy is set above the main absorption peak. In particular, the low T spectrum is stronger than the high temperature one. Interestingly, the effect is much smaller if not zero at the main absorption peak, both at Cu L 3 and at O k edges. Despite the large number of cuprate RIXS spectra published in the last decade, this effect has never been discussed before. The authors attribute the effect to the manifestation, on a relatively high energy scale, of the changes in the electronic structure due to the opening of the superconducting gap. Which seems a very reasonable assumption. Moreover, they were guided in this experiment by the theoretical predictions of reference 26, so they naturally conclude that the experiment can be seen as the confirmation of the TCFM picture. The latter appears to be more of a speculation than a demonstration.

The merit of the work is to unveil the T dependence of the RIXS excitonic spectral region when properly excited by choosing the photon energy corresponding, in the XAS, to the “hole doping” states, ie about 1 eV above the main peak. This is potentially very important, as it provides a new playground for RIXS studies of the superconducting, pseudogap and strange metal states of cuprates with higher sensitivity than in the past. However, the manuscript presents only 2 temperatures and the connection of the effect with the superconducting transition is not demonstrated. The comparison with the calculated spectra is rather deceiving. The sign of the effect is indeed the same (stronger intensity at low T) and the importance of exciting above the absorption peak is also found in the calculations, but the spectral shape is dramatically different. This is not surprising given that the calculations are for just a fraction of the possible set of final states of the RIXS process: spin and orbital degrees of freedom are missing in the intermediate energy range, as well as phonons and charge order in the low energy scale. The RIXS cross sections are oversimplified, the intermediate state is approximated

by a bare core hole potential, and it is not clear if the experimental geometry is taken in account in any ways. Therefore, it is not surprising that the comparison does not lead to a significant agreement besides the correct sign of the difference between the two temperatures.

The novelty of the results is in the experimental spectra. The theoretical model, its hypotheses and implications, had been published and discussed before. The question is whether the spectra measured on Bi2212 and presented here are providing a significant support to the theoretical picture. The maximum one can say is that the experiment is not totally incompatible with the theory. But one would need a more extensive comparison to convincingly support the TCFM model and its numerous implications about the physics of cuprates. Several points are missing that would make the connection between theory and experiment more convincing:

- Both calculations and measurements deal with 2 temperatures only. The gap and the pseudogap temperatures should be crossed with finer steps in temperature to know what the model predicts and if it is in agreement with the experimental findings.

- The low energy scale, below 100 meV, is neglected. If one can understand that here the experimental resolution is not adequate to study the opening of the superconducting gap as in reference 44, it is surprising that the theoretical spectra of figures 3 and 4 disregard completely the low energy scale.

- Also the "high" energy scale is somehow forgotten. The experiment shows that the largest effects are on the dd excitation peak, between 1 and 2.5 eV. The discussion on this observation is very limited, and the phenomenon is attributed to a sort of shake-up process, not included in the model. A more elaborated discussion of this finding is necessary.

In conclusion, I think that the manuscript is reporting a potentially interesting observation on the RIXS of cuprates but the claims of having confirmed the TCFM are largely unsubstantiated, because the experimental basis is too narrow and the agreement with theoretical spectra is only qualitative. These results are indeed interesting to the RIXS community and can be published in a more specialized journal in the present form. To deserve publication in Nature Communications a more extensive and quantitative agreement with the calculations should be shown.

Summary of changes

In response to the referees' comments, we have revised the manuscript to improve the presentation of our results. Major changes made in the revised manuscript are highlighted in blue. We list the major changes in the following.

- 1) Figures 1 and 2 in the original manuscript have been combined into new Fig. 1.
- 2) Two new figures—RIXS on an overdoped sample (new Fig 3) and RIXS of temperature changes across the superconducting transition (new Fig 4)—have been included.
- 3) Two new paragraphs—4th and 5th paragraphs—have been added to the introduction section.
- 4) The 3rd paragraph of section Excitons revealed by Cu L-edge RIXS has been greatly revised.
- 5) Two new paragraphs—2nd and 3rd paragraphs—have been added to section Suppression of excitons in the pseudogap phase.
- 6) The 1st paragraph of section Two-component fermion model analysis has been revised in depth.
- 6) In section Discussion and Conclusion, two paragraphs –1st and 2nd paragraphs—have been added.
- 7) In section Methods, the explanations on TCFM have been extensively revised.
- 8) Seven new references (Refs. 12, 23, 37, 39, 43, 60, 61) have been cited.

Reply to Referees

We greatly thank you for your valuable remarks, which led us to improve the quality of the manuscript significantly. In response to your comments, we have thoroughly revised the manuscript. The major revised parts are highlighted in blue in the main text. In the following, we reply to the referees' remarks item by item in detail. We also revised the manuscript thoroughly to enhance the readability without changing the content and to correct typos in a number of places.

Reviewers' comments and our replies:

Comments of Reviewer #1:

The authors of the manuscript by Singh et al. aim to study the exciton dynamics originating from electron fractionalization of a superconducting cuprate into the lower and upper Hubbard bands. Resonant Inelastic X-Ray Scattering (RIXS) experiments at Cu L3 and O K-edges on optimally hole doped $\text{Bi}_2\text{Sr}_2\text{CaCu}_2\text{O}_{8+\delta}$ (Bi-2212) display a medium-energy response around 1eV framed as excitonic excitations much beyond the superconducting-gap energy scale. The main experimental finding of the study is that these excitations are unusually enhanced upon entering the superconducting state. The authors see this as a clear proof of the involvement of such excitons in the superconductive mechanism.

The experimental RIXS data presented in the paper are certainly of high quality and the observation of such a spectral weight enhancement across the superconducting transition is certainly interesting. The experimental test of a hypothesis made from a state-of-the-art model calculation is an original result. I find that the topic could in principle be suitable for high profile journals like Nature Communications. However, the paper largely describes an experimental test of the prediction of inelastic scattering enhancement made in already published theory papers (ref. 23 and 26) by a simple two-component fermion model, which is based on the electron fractionalization concept. By this only part of the paper is fully original. For instance, Fig. 1 is recycled from the previous publication. A further crucial shortcoming of the manuscript is that the physical picture suggested in ref. 23 and 26 is described in a way that is not appropriate for the general audience. It would be required for a publication in Nature Communications that a broad audience can follow the proposed physical concept without consulting several theory papers delivering the background. The referee finds the concept impossible to follow and comprehend without consulting these previous publications. Coming to the experimental results, the description of the incident energy dependence of the RIXS data in Fig. 2 is very difficult to follow as well. The development of the magnon excitations with a Raman response around the Cu L3 resonance into a broad exciton showing a fluorescence behavior is very bad explained. Apparently, this invokes an electron-hole pair mechanism that would need to be described in detail. For a broader audience a schematic illustration of such scattering mechanism could possibly illuminate

such complicated processes. The authors seem to claim that the enhanced spectral response must be due to charge scattering. The argument for this lies in an ad hoc explanation that the changing spectral weight must be of charge origin, since the 0.5 eV magnon response does not show any spectral enhancement with temperature. This argument is inconclusive, as indeed the response around 0.5 eV energy loss is known to be composed of both spin and charge excitations, when no polarization selectivity is applied in the detected channel. Looking how well the experimental data agree with the calculations in Fig. 3 is also not easy, as these panels present a zoom into different energy scales. At best, there is a qualitative agreement. The authors claim an enhancement in the order of 5-10%, but do not show any quantitative analysis. Actually, the clearer temperature development is seen not in the exciton region around 1 eV, but rather in the spectral region of the dd-excitations above 2 eV, which seems to be much more important than the region around 1 eV. Unfortunately, the model does not describe this spectral region. Therefore, the claimed “stringent test of a theoretical prediction” is by no means unambiguous.

In summary, the paper reports interesting medium-energy scattering data by RIXS across the superconducting transition that are not conclusively analyzed with a previously developed theory model. The paper lacks the urgency required by Nature Communications and should better be submitted to a more specialized journal. This will require, however, a total rewrite of the paper that explains the processes and spectral development much more clearly.

Reply:

We first thank the referee for his/her valuable comments. We appreciate that the referee agrees that the manuscript is of high quality and that the observation is interesting. In the following, we reply to the referee’s comments point-by-point: (Referee’s comments are repeated in blue.)

(1) Fig. 1 is recycled from the previous publication.

Reply:

The central result of our work is that the exciton intensity is enhanced from the PG phase to the SC phase. To explain this unusual phenomenon, we need to share the basic knowledge of the electronic structure known for the cuprates with readers. For this purpose, it is reasonable to use the results already established in the literature to show this background as an introduction for general readership. This is indeed necessary as a premise when we discuss our main result of the enhancement in the following sense: Previous Fig. 1a, which is reproduced from Ref. [28] (Charlebois et al, PRX 10, 041023 (2020)), clearly showed the dispersions of the in-gap band (IGB) and the coherent lower Hubbard band (LHB) with realistic parameter values indispensable for the present purpose of energy-scale comparison. In response to the referee’s comment, we have merged previous Figs. 1 and 2 into one new Fig. 1, which

makes it possible to understand the one-to-one correspondence between the exciton peak in the RIXS data (Fig.1b) and the basic single-particle excitation spectra (Fig.1a). We have addressed this correspondence in the caption of Fig.1.

(2) A further crucial shortcoming of the manuscript is that the physical picture suggested in ref. 23 and 26 is described in a way that is not appropriate for the general audience. It would be required for a publication in Nature Communications that a broad audience can follow the proposed physical concept without consulting several theory papers delivering the background. The referee finds the concept impossible to follow and comprehend without consulting these previous publications.

Reply:

In the original manuscript, we described the introduction to the TCFM only briefly to avoid repetition of the literature, because the TCFM theory itself is not the original part of the present paper and is already discussed in detail in the literature. However, in response to the referee's criticism "the referee finds the concept impossible to follow and comprehend without consulting these previous publications", we have revised the description of TCFM substantially. First in the introduction, second in section Two-component fermion model analysis (page 5), third in Method section, and finally in Supplementary material; we believe, now it is easy to follow and understand TCFM by starting from an intuitive picture and with a detailed but compact review of the properties of TCFM. In Introduction, we have additionally addressed the following points by preparing the new 4th and 5th paragraphs:

"Electron fractionalization also embodies the idea of the duality of strongly correlated electrons. This duality is represented by, on the one side, the conventional itinerant quasiparticles localized in momentum space connected with the overdoped Fermi liquid. The other side is represented by electrons localized in real space leading eventually to the Mott insulator in the undoped limit. Such a duality shares a common concept of the coexistence of itinerant and localized electrons proposed in the literature for metals, for instance, in Refs. [1, 23]. As for the localized nature of the electron, as is known in the Mott insulator, an electron is bound to a hole, requiring a charge gap to split into an electron and a hole. Such an electron bound to a hole is charge neutral in total and does not primarily interact with the electromagnetic wave. This binding may remain in the carrier doped case.

A two-component fermion model (TCFM), which will be detailed later in the present article, was proposed to embody such a dual character of electrons. The dark fermion called "d fermion," one of the two constituents of the TCFM, represents the localized side of the electron bound to a hole induced by doping. The dark fermion is expected to be more stable near the Mott insulator, while the other, called "c fermion," characterizes the coherent metallic component of electrons and becomes

more stable in the overdoped region. The TCFM manifests this bistable nature of electrons in lightly carrier-doped systems.”

In the Two-component fermion model analysis section starting from page 5, we further deepen this intuitive view and make the TCFM Hamiltonian easily understandable by pointing out that the two-component c and d fermions playing a role of the bistable character of electrons are hybridizing each other. This part was already in the original manuscript, but now the descriptions added to the introduction (4th and 5th paragraphs) make the access to this part easier. Through the quantum mechanical hybridization of these two constituents, the formation of bonding and antibonding states naturally gives rise to the pseudogap in TCFM.

In the 2nd paragraph of the TCFM section, we use the nature of a dark fermion—non-interacting with the electromagnetic waves—to explain the exciton enhancement from the PG phase to the SC phase. We have added further details by introducing the explicit form of the Hamiltonian given by the itinerant c fermion and nearly localized d fermion (dark fermion) in the second paragraph of Methods. Quantitative estimates of the Hamiltonian parameters and its physical properties have also been added to help the understanding of this paper. We have also added a statement on how the TCFM parameters are chosen to quantitatively reproduce the ARPES data in Supplementary Information with the heading “Two-component fermion model”.

(3) The description of the incident energy dependence of the RIXS data in Fig. 2 is very difficult to follow as well. The development of the magnon excitations with a Raman response around the Cu L3 resonance into a broad exciton showing a fluorescence behavior is very bad explained....The authors seem to claim that the enhanced spectral response must be due to charge scattering. The argument for this lies in an ad hoc explanation that the changing spectral weight must be of charge origin, since the 0.5 eV magnon response does not show any spectral enhancement with temperature.

Reply:

For the soft X-ray RIXS of cuprates, there are extensive studies showing that the paramagnon excitation has an energy below 0.5 eV, the peak position of which depends on the momentum transfer but not on the incident energy; see Refs. [32-38]. In addition, the observed 0.5-eV feature in RIXS contains the electron-hole-pair excitation, which shows a fluorescence-like RIXS feature and a shift with excitation energy when the incident energy is varied. This assignment has been well established both experimentally and theoretically. We have cited two more references, Refs. [37,39], and revised the third paragraph of section Excitons revealed by Cu L-edge RIXS on page 3 to respond to the referee’s comment that the argument lies in an ad hoc explanation. We added the statement “*As the X-ray energy is increased above*

$L_3+0.6$ eV, this RIXS feature appears to shift toward high energy loss, and evolves into a fluorescence-like excitation. The observed fluorescence-like shift is due primarily to the continuum of particle-hole excitation in the charge channel [33,34,39], i.e., an exciton excitation created by the transition of an electron in the Cu 2p core level to the IGB (Fig.1a, left panel), followed by the transition of an electron in the coherent LHB to the Cu 2p core level (Fig.1a, right panel).” to lines 152-161 in the third paragraph of section Excitons revealed by Cu L-edge RIXS, and “The photon energy ω determines the energy ω_0 of the excited electron above the Fermi level, i.e., $\omega = E_F + \omega_0$.” to lines 176-177 in the fourth paragraph of section Excitons revealed by Cu L-edge RIXS. In addition, to strengthen our assignment that the excitation in this energy range is charge channel, we added “The assignment of the peak structure in this energy range to the charge excitation (namely exciton formation) is corroborated by the fact that O K-edge spectra shown later also exhibits a similar corresponding peak, indicating that the observed structure arises from the charge channel as the single-spin-flip process is forbidden in the O K-edge RIXS.” to lines 161-167 in the third paragraph of section Excitons revealed by Cu L-edge RIXS. On page 5, we added “Note that the enhancement observed both in Cu L_3 - and O K-edges evidences that it is ascribed to the excitonic charge channel and not to the single-spin flip magnetic excitations.” to lines 294-297 in section Suppression of excitons in the pseudogap phase.

(4) How well the experimental data agree with the calculations in Fig. 3 is also not easy, as these panels present a zoom into different energy scales. At best, there is a qualitative agreement. The authors claim an enhancement in the order of 5-10%, but do not show any quantitative analysis.

Reply:

Although the experimental and calculated line shapes around 1 eV look very different at first glance (Figs. 2 and 5), the difference can be simply attributed to the strong tail of the well established dd excitation peak at 2-3 eV and other high-energy excitation, and the paramagnon excitation at < 0.5 eV. After subtracting them out, one can clearly identify the peak structure around 1 eV that can be attributed to the exciton and can be compared with the TCFM prediction. Bearing this in mind, we find remarkable quantitative agreement between the experiment and calculation in the following sense:

(a) The excitonic peak energy in the RIXS spectra has an overall and quantitative agreement between the experiment and calculation: For instance, for the Cu L_3 edge at momentum transfer $(\pi,0)$, the calculated peak energy ω_{peak} is approximately equal to the incident energy ω_0 , measured from the L_3 peak or the absorption threshold, in

both experiment and calculation. The peak width is roughly 0.5-1.0 eV, which is also in agreement between experiment and calculation.

(b) The calculated enhancement of the exciton peak in the superconducting phase is 7.6% for the Cu L_3 edge at $\omega_0=1.2$ eV. In the case of the experiment, as one sees in Fig. 4c, the enhancement at temperatures far below T_c from the pseudogap region around 110K-150K is around 7-10%, again in good agreement between the experiment and calculation.

In the above comparisons, the parameters of the TCFM model are solely chosen to reproduce the ARPES data without further adjustment. Because ARPES data do not contain the unoccupied part, TCFM parameters may have larger errors for those related to the unoccupied degrees of freedom. Nevertheless, we have obtained good agreement between the TCFM calculation and RIXS data for the exciton energy (a) and the enhancement ratio (b).

By considering the simplicity of TCFM without parameter adjustment, this quantitative agreement is highly nontrivial. RIXS data contain richer information about the unoccupied part of the single-particle spectra as well as the two-particle exciton spectra. The present basic agreement assures that integrated analyses of ARPES and RIXS together with other spectroscopic data will provide us with more comprehensive understanding in the future. We have added the above statements in the first and subsequent paragraphs of section Discussion and Conclusion (see lines 376-416, pages 6-7).

(5) The clearer temperature development is seen not in the exciton region around 1 eV, but rather in the spectral region of the dd -excitations above 2 eV, which seems to be much more important than the region around 1 eV. Unfortunately, the model does not describe this spectral region.

Reply:

We have focused on the exciton formed between the IGB and the coherent LHB for comparison with TCFM calculation, where the high energy component is not considered. Within the framework of this low-energy effective model, we have explained the temperature dependence of the dd -excitation region by a shake-up mechanism. The temperature dependence of the dd region may look as strong as that in the 1 eV region, but this is not the case, as evidenced by the new intensity normalization over a wide energy range (Figs. R1 and R2 in the reply to Referee #2). On the theoretical side, the extension of the TCFM to include the upper Hubbard band requires a substantial extension of the model. The parameters needed for the extension are not available so far from the ARPES data either. Therefore, this is not within the scope of this paper and is left for future studies. However, the physics of TCFM will equally apply to an exciton with an upper Hubbard band electron, and a

similar enhancement is expected. We do not know any other mechanism that explains the similar enhancement in these two energy regions. The enhancement at least at the IGB exciton region is by itself worth attracting the attention of the wide readership. As an intuitive explanation of why similar physics is expected in the high energy region, we have introduced a “shake-up mechanism” in Discussion and Conclusion (see lines 440-449, page 7), which was moved from the last paragraph of Two-component fermion model analysis section and was elaborated in the revised manuscript to meet the referee’s request.

(6) In summary, the paper reports interesting medium-energy scattering data by RIXS across the superconducting transition that are not conclusively analyzed with a previously developed theory model. The paper lacks the urgency required by Nature Communications and should better be submitted to a more specialized journal. This will require, however, a total rewrite of the paper that explains the processes and spectral development much more clearly.

Reply:

The mechanism of high-temperature superconductivity in cuprates has long remained a mystery for decades. In particular, the underlying physics of the pseudogap phase remains unclear. Here, we present unprecedented two-particle spectroscopic results, which show an unusual enhancement of excitonic excitation caused by superconductivity. To the best of our knowledge, this atypical change in the spectral feature of excitons can only be explained by a novel theoretical idea based on electron fractionalization but also calls for other ways, if possible, to deepen the understanding of these unexpected remarkable results. We believe the combined experimental and theoretical results will lead to a paradigm shift in understanding the pseudogap phase.

As per Referee requested, we have significantly modified the manuscript to improve our presentation. We hope the referee shares this viewpoint and agrees with the publication of our manuscript. We thank the referee again for the valuable remarks, which led us to improve the quality of the manuscript.

Comments of Reviewer #2:

In the manuscript by Singh et al., the authors report on a resonant inelastic X-ray scattering (RIXS) experiment carried out on an optimally-doped Bi2212 single crystal. Their central finding is made when the incident photon energy is detuned to the higher-energy side of copper L3 and oxygen K absorption edges by about 1 eV, where their experimental configuration (sigma-polarization grazing incident) is known to allow for observations of charge excitations (which the authors call excitons) arising from the generation of particle-hole pairs.

The claimed novel observation is that the intensity of the signal, primarily of the fluorescence (rather than the 'Raman') type, is stronger at $T = 50$ K than at 120 K, the former of which is below the superconducting critical temperature T_c of the sample. It is argued that this below- T_c enhancement is caused by a competition between superconductivity and pseudogap physics, the latter of which is presumed to cause quasiparticle fractionalization, such that when superconductivity helps restore the quasiparticle component, the RIXS intensity is enhanced. It is concluded that the observed temperature-dependent signals at relatively large nominal momentum transfers have been experimentally missed so far, and that the phenomenon proves the involvement of high-energy excitons in superconductivity; this in turn supports the so-called two-component fermion model, in which the pseudogap phase is theoretically known to feature fractionalized quasiparticles.

The paper is well-written, with a clear motivation for the study and smooth logic flow. I also tend to agree that if superconductivity can indeed enhance a new type of resonant scattering (or fluorescence) intensity, at a much higher energy than the pairing gap and with peculiar characteristics allowing for pin-pointing the underlying mechanism to be within the two-component fermion model (or even just any models that ensures quasiparticle fractionalization), it would be a result novel enough to warrant publication in Nature Communications. Here, in agreement with the authors' own claim, I consider the peculiar characteristics to include (i) prominence of the effect beyond physically trivial interpretations, (ii) observable only at high momentum transfers, (iii) enhancement below T_c , and (iv) linkage to the pseudogap phase and its competition with the superconductivity. Unfortunately, I see a series of difficulties with essentially all of these aspects, and remain completely unconvinced of the authors' claim.

1) The authors use a normalization method to compare data sets below and above T_c , as explained in Fig. 3 captions: the normalization keeps intensities at larger than 5 eV energy loss constant. This procedure leads to a very suspicious consequence, in that the d-d excitation peak area changes considerably, under precisely the same detuned incident energies where the largest enhancement (below T_c) is seen for the 'excitons'. An alternative way of normalization, which is quite commonly employed in Cu L3 RIXS experiments, is to use the d-d area (keeping the d-d peak area constant below and above T_c). What would the authors get if they do this instead? I am particularly concerned by the fact that their displayed RIXS energy spectra are only up to ~ 6.5 eV, and the integrated spectral weight between 5 eV and 6.5 eV is only a small fraction of the total intensity between 0 and 6.5 eV. Then, a small normalization systematic error could result in a large bias on the important peaks' (both d-d and exciton) intensities. Moreover, it is well known that RIXS intensity needs to be corrected for sample's self-absorption (for the scattered photons), and the strength of such absorption can be strongly energy-loss-dependent too, especially if the incident energy is detuned to the high-E side. Since the authors' conclusion heavily depends on spectral comparison at a detailed level, and because the observation involves scattered photons that are on-resonance (detuned by ~ 1 eV, then with ~ 1 eV energy loss), these technical aspects along with the uncertainties they bring need to be handled with special care, but I do not find them

being carefully discussed at all. For me, this severely hampers the credibility of the entire pack of the experimental results.

The authors made an argument related to shake-off effects to explain the change in the d-d peak with temperature, which makes no sense to me at all: With such shake-off, the d-d peak enhancement should not be restricted to only the 'correct' incident photon energies. To me it looks very much like a systematic error with the normalization procedure. Please check.

Reply:

We thank the referee for this comment and agree that a small normalization systematic error could result in a bias in the conclusion. We first apologize for the confusion about the normalization range. We measured RIXS spectra up to 13 eV energy loss and the normalization was done for energy loss higher than 4 eV. In the original manuscript, we plotted RIXS spectra up to 6.5 eV for representation only, which has been changed up to 12.5 eV in the revised manuscript. Figures R1 and R2 attached at the end of the reply present a comparison between the two normalization ranges, one covering the *dd* peak only and the other covering the *dd* peak and its high-energy tail. The comparison shows that both schemes give rise to consistent results—the exciton is enhanced as the temperature is changed from the PG phase to the SC phase. That is, our conclusion does not result from a small normalization systematic error. In the revised manuscript, the spectra were normalized to the integration between 1.7 and 13 eV.

Thanks to the referee for pointing out the self-absorption effect. We agree that the energy-loss spectra can be strongly self-absorption dependent. However, our focus is on the comparison of the change between the PG and SC phases. For a given energy loss and a momentum transfer, the RIXS intensity difference between two temperatures is not affected by the self-absorption effect. That was the reason we presented the comparison between spectra of two temperatures without correction for self-absorption. In the revised manuscript, we wrote a note in the caption of Fig. 2 saying that the self-absorption effect was not corrected.

As for the shake-up effect, it refers to the excitation of the photo-excited electron from the IGB to the UHB, arising from an effect of the Cu *2p* core-hole potential. We have discussed the enhancement of the UHB-coherent LHB exciton peak and not the *dd* peak. The *2p* core-hole potential pulls down the UHB in the intermediate state of RIXS. Therefore, the photo-excited electron has a finite probability of entering the UHB in addition to entering the IGB. As these descriptions were missing in the original manuscript, we have included them in the revised manuscript. In response to the referee's comments, we have revised the fourth paragraph of the Discussion and Conclusion section to elaborate on the shake-up effect (see lines 440-449, page 7).

2) The authors studied only two momenta ($\pi,0$) and ($\pi/2,0$), and at both places they found an enhancement. To really show that the effect pertains to large Q transfer only, and to make a case consistent with previous studies reporting no such effects, I expect the authors to show a much weaker effect or absence of effects at smaller Q using their same sample, same experimental set up.

Reply:

To observe the enhancement effect, we need to measure excitons that can be easily separated from paramagnon and plasmon excitations. However, at low Q, the IGB-coh LHB exciton has a low energy and will more heavily overlap with paramagnons, which have also low energies at low Q. Also, acoustic plasmons will contribute to the low-Q RIXS spectrum and might smear out the observation of the enhancement effect. To present convincing experimental evidence for the exciton enhancement, we therefore decided to focus on high-Q measurements at more temperatures across the superconducting transition.

3) Data at only two temperatures are insufficient for claim a connection to T_c . If the authors really believes in such a connection, they should at least be able to show a full variable-T data set by focusing on just one Q and one good incident energy. Being able to show such a data set would also significantly help resolve my point 1) above, because when data at many temperatures show that only the exciton intensity is strongly changing near T_c , it would be unlikely to be due to systematic errors related to normalization.

Reply:

In response to the referee's comment, we have measured more temperatures across T_c on OP Pb-Bi2212. Figure 4c in the revised manuscript shows first a decrease from 250 to 150 K, where the pseudogap grows. Then further decrease of temperature below the superconducting fluctuation temperature $T_{scf} \sim 110$ K shows a sharp increase of the intensity to the saturated value in the superconducting phase below T_c . We can, therefore, clearly conclude that the observed enhancement is not due to systematic errors related to normalization. On the contrary, we have succeeded in observing the suppression of the RIXS intensity in the pseudogap phase and its recovery in the superconducting phase in perfect agreement with the prediction of TCFM. See lines 236-248 in section Suppression of excitons in the pseudogap phase for discussions on these new data.

4) I know RIXS is a demanding type of experiment, but if the claimed novel phenomenon does require a competition between superconductivity and pseudogap phase, studying an overdoped sample (and observing absence of similar effects) would be a very good bench-marking experiment. It is also well-known (see, e.g., Minola et al., PRL 119, 097001

(2017), including supplementary materials Fig. 2) that the charge excitations are more prominent on the overdoped side, so the study should at least be feasible.

Reply:

In response to the referee's comment, we have measured RIXS on an OD sample without a clear signature of the pseudogap. Figure 3 in the revised manuscript displays Cu L_3 -edge RIXS of OD Pb-Bi2212 at temperatures above and below T_c . We observed no enhancement induced by the superconductivity in the exciton spectral weight of the OD Pb-Bi2212 sample. This leads us to conclude that the observed exciton enhancement in the superconducting state (or conversely the exciton suppression in the pseudogap state) in OP Bi2212 is connected to the existence of the pseudogap phase.

In summary, I am rather unconvinced of the technical correctness (primarily experimentally) of the results presented in their present form. I believe that the above issues need to be overcome before the work can be published.

Reply:

We believe that the above point-by-point replies are satisfactory. We particularly thank the referee for bringing us to the two crucial measurements—RIXS data of an over-doped sample for a benchmark and detailed temperature-dependent measurements across T_c . We have successfully carried out those two measurements. The results fully support the results drawn in the original manuscript. Now we hope the referee agrees that the results and the conclusion of the revised manuscript are convincing. We thank the referee again for the valuable remarks, which led us to improve the quality of the manuscript.

Figure R1: RIXS spectra at selected incident photon energies for $Q_{||} = (\pi, 0)$ at temperatures above and below $T_c = 89$ K. Left panel: the spectra are normalized to the dd excitation at energy loss (1.6-4.8 eV). Right panel: the spectra are normalized to the integrated intensity in the 1.7- 13 eV energy loss range.

Figure R2: Zoom-in RIXS spectra at selected incident photon energies for $Q_{||} = (\pi, 0)$ and temperatures above and below $T_c = 89$ K. Left panel: the spectra are normalized to the dd excitation at energy loss (1.6-4.8 eV). Right panel: the spectra are normalized to the integrated intensity in the 1.7- 13 eV energy loss range.

Comments of Reviewer #3:

The manuscript presents the comparison of RIXS spectra of Bi2212 measured across the Cu L3 and O K edges with theoretical spectra of the particle-hole channel calculated with the “two-component fermion model” (TCFM). Whereas the latter results were introduced recently by one of the authors in reference 26, although for a limited number of cases, the former are more original. The main observation is that RIXS spectra measured at 50K and 120K, ie below and above the superconducting transition temperature of the chosen sample, visibly differ in intensity in the 0.3-2.5 eV energy loss range when the incident photon energy is set above the main absorption peak. In particular, the low T spectrum is stronger than the high temperature one. Interestingly, the effect is much smaller if not zero at the main absorption peak, both at Cu L 3 and at O k edges. Despite the large number of cuprate RIXS spectra published in the last decade, this effect has never been discussed before. The

authors attribute the effect to the manifestation, on a relatively high energy scale, of the changes in the electronic structure due to the opening of the superconducting gap. Which seems a very reasonable assumption. Moreover, they were guided in this experiment by the theoretical predictions of reference 26, so they naturally conclude that the experiment can be seen as the confirmation of the TCFM picture. The latter appears to be more of a speculation than a demonstration.

The merit of the work is to unveil the T dependence of the RIXS excitonic spectral region when properly excited by choosing the photon energy corresponding, in the XAS, to the “hole doping” states, ie about 1 eV above the main peak. This is potentially very important, as it provides a new playground for RIXS studies of the superconducting, pseudogap and strange metal states of cuprates with higher sensitivity than in the past. However, the manuscript presents only 2 temperatures and the connection of the effect with the superconducting transition is not demonstrated. The comparison with the calculated spectra is rather deceiving. The sign of the effect is indeed the same (stronger intensity at low T) and the importance of exciting above the absorption peak is also found in the calculations, but the spectral shape is dramatically different. This is not surprising given that the calculations are for just a fraction of the possible set of final states of the RIXS process: spin and orbital degrees of freedom are missing in the intermediate energy range, as well as phonons and charge order in the low energy scale. The RIXS cross sections are oversimplified, the intermediate state is approximated by a bare core hole potential, and it is not clear if the experimental geometry is taken in account in any ways. Therefore, it is not surprising that the comparison does not lead to a significant agreement besides the correct sign of the difference between the two temperatures.

The novelty of the results is in the experimental spectra. The theoretical model, its hypotheses and implications, had been published and discussed before. The question is whether the spectra measured on Bi2212 and presented here are providing a significant support to the theoretical picture. The maximum one can say is that the experiment is not

totally incompatible with the theory. But one would need a more extensive comparison to convincingly support the TCFM model and its numerous implications about the physics of cuprates. Several points are missing that would make the connection between theory and experiment more convincing:

- Both calculations and measurements deal with 2 temperatures only. The gap and the pseudogap temperatures should be crossed with finer steps in temperature to know what the model predicts and if it is in agreement with the experimental findings.

- The low energy scale, below 100 meV, is neglected. If one can understand that here the experimental resolution is not adequate to study the opening of the superconducting gap as in reference 44, it is surprising that the theoretical spectra of figures 3 and 4 disregard completely the low energy scale.

- Also the “high” energy scale is somehow forgotten. The experiment shows that the largest effects are on the dd excitation peak, between 1 and 2.5 eV. The discussion on this observation is very limited, and the phenomenon is attributed to a sort of shale-up process, not included in the model. A more elaborated discussion of this finding is necessary.

In conclusion, I think that the manuscript is reporting a potentially interesting observation on the RIXS of cuprates but the claims of having confirmed the TCFM are largely unsubstantiated, because the experimental basis is too narrow and the agreement with theoretical spectra is only qualitative. These results are indeed interesting to the RIXS community and can be published in a more specialized journal in the present form. To deserve publication in Nature Communications a more extensive and quantitative agreement with the calculations should be shown.

Reply:

We first thank the referee for his/her valuable comments. We appreciate that the referee agrees that the manuscript is of high quality and that the observation is interesting. In the following, we reply to the referee’s comments point-by-point: (Referee’s comments are repeated in blue).

–The comparison with the calculated spectra is rather deceiving. ... the spectral shape is dramatically different. This is not surprising given that the calculations are for just a fraction of the possible set of final states of the RIXS process...

Reply:

We respectfully disagree with the statement “The comparison with the calculated spectra is rather deceiving”. Although the experimental and calculated line

shapes around 1 eV look very different at first glance (Figs. 2 and 5), the difference can be simply attributed to the strong tail of the well established dd excitation peak at 2-3 eV and other high-energy excitation, and the paramagnon excitation at < 0.5 eV. After subtracting them out, one can clearly identify the peak structure around 1 eV that can be attributed to the exciton and can be compared with the TCFM prediction. Bearing this in mind, we find remarkable quantitative agreement between the experiment and calculation in the following sense:

- (a) The excitonic peak energy in the RIXS spectra has an overall and quantitative agreement between the experiment and calculation: For instance, for the Cu L_3 edge at momentum transfer $(\pi,0)$, the calculated peak energy ω_{peak} is approximately equal to the incident energy ω_0 , measured from the L_3 peak or the absorption threshold, in both experiment and calculation. The peak width is roughly 0.5-1.0 eV, which is also in agreement between experiment and calculation.
- (b) The calculated enhancement of the exciton peak in the superconducting phase is 7.6% for the Cu L_3 edge at $\omega_0=1.2$ eV. In the case of the experiment, as one sees in Fig. 4c, the enhancement at temperatures far below T_c from the pseudogap region around 110-150 K is around 7-10%, again in good agreement between the experiment and calculation.

In the above comparisons, the parameters of the TCFM model are solely chosen to reproduce the ARPES data without further adjustment. Because ARPES data do not contain the unoccupied part, TCFM parameters may have larger errors for those related to the unoccupied degrees of freedom. Nevertheless, we have obtained good agreement between the TCFM calculation and RIXS data for the exciton energy (a) and the enhancement ratio (b).

By considering the simplicity of TCFM without parameter adjustment, this quantitative agreement is highly nontrivial. RIXS data contain richer information about the unoccupied part of the single-particle spectra as well as the two-particle exciton spectra. The present basic agreement assures that integrated analyses of ARPES and RIXS together with other spectroscopic data will provide us with more comprehensive understanding in the future. We have added the above statements in the first and subsequent paragraphs of Discussion and Conclusion (see lines 376-416, pages 6-7).

– The RIXS cross sections are oversimplified, the intermediate state is approximated by a bare core hole potential,

Reply:

For our experimental geometry with σ -polarized incident X-rays, charge excitation without spin flip is dominant and the RIXS matrix element can be treated approximately constant. In particular, for the TCFM calculation, which only treats the

charge degree of freedom, one can directly compare the calculation with our RIXS spectra. The purpose of the comparison with the TCFM is to extract the essence of the enhancement mechanism in the complex experimental setup. If we could treat the “theory of everything” with full consideration of the details, though in reality it is not possible, we would miss to extract the essence and fail to identify the mechanism. To understand the mechanism intuitively, we need to resort to a skeletonized model. If a simplified model explains the main issue of the enhancement, that would give us deep insight. Since the model is simple, the agreement in terms of the enhancement with the TCFM in this sense provides us with a clear understanding about the essence of the mechanism.

– It is not clear if the experimental geometry is taken into account in any ways.

Reply:

The experimental X-ray polarization is along one of the Cu-Cu bond directions in the CuO_2 plane, which is a right orientation in the model of calculation because the matrix element c in the RIXS intensity formula Eqs. (4) and (5) derived from the dipole moment is nonzero and (4) and (5) are an adequate formula for the RIXS intensity. Other setup in the energy momentum does not have any conflict. We have added this statement after Eq. (5).

- Both calculations and measurements deal with 2 temperatures only. The gap and the pseudogap temperature should be crossed with finer steps in temperature to know what the model predicts and if it is in agreement with the experimental findings.

Reply:

We thank the referee for his/her valuable comments. In response to this comment, we have measured more temperatures across T_c on another crystal, Pb-Bi2212. Figure 4c shows first a decrease from 250 to 150 K, where the pseudogap grows. Then further decrease of temperature below the superconducting fluctuation temperature $T_{\text{scf}} \sim 110$ K shows a sharp increase of the intensity to the saturated value in the superconducting phase below T_c .

This detailed temperature dependence demonstrates that the pseudogap generates the suppression of the RIXS intensity and the superconducting transition removes this suppression in agreement with the fractionalization picture. See lines 236-248 in section Suppression of excitons in the pseudogap phase for discussions on these new data.

- The low energy scale, below 100 meV, is neglected. If one can understand that here the experimental resolution is not adequate to study the opening of the superconducting gap as

in reference 44, it is surprising that the theoretical spectra of figures 3 and 4 disregard completely the low energy scale.

Reply:

Thanks to the referee for understanding that our Cu *L*-edge RIXS does not have a good energy resolution to measure the SC gap. For theoretical spectra, we have employed the parameters of the TCFM chosen to reproduce the ARPES data (Ref. 59) together with those analyzed by machine learning (Ref.24) and STM data (See Supplementary Information). These parameters indeed well fit the ARPES data and the low-energy self-energy structures including those obtained from the machine learning analysis, as shown in Fig. S8a, c, d in Supplementary Information. For instance, the well-known peak-dip hump structure of the spectral weight for Bi2212 in the superconducting phase is quantitatively reproduced as is shown in supplementary Fig. S7a on the low- and high energy scales in the measured range of ARPES. On the other hand, the effect of *d*-wave superconductivity on RIXS spectra is negligibly small at low energies at large momentum transfer such as $(\pi,0)$ and $(\pi/2,0)$ in TCFM. See, e.g., Fig. 9 of Ref. [26]. This negligible effect of superconductivity can be understood from the fact that the energy range of the hole forming the exciton is already gapped in the pseudogap state, where the full gap opens. The contribution from the nodal region does neither show a change between the superconducting state and the pseudogap state because the electronic state does not change in the nodal region.

- Also the “high” energy scale is somehow forgotten. The experiment shows that the largest effects are on the dd excitation peak, between 1 and 2.5 eV. The discussion on this observation is very limited, and the phenomenon is attributed to a sort of shake-up process, not included in the model. A more elaborated discussion of this finding is necessary.

Reply:

We have revised the Discussion and Conclusion section of the revised manuscript to elaborate on the shake-up effect. See lines 442-451 on page 7.

In conclusion, I think that the manuscript is reporting a potentially interesting observation on the RIXS of cuprates but the claims of having confirmed the TCFM are largely unsubstantiated, because the experimental basis is too narrow and the agreement with theoretical spectra is only qualitative. These results are indeed interesting to the RIXS community and can be published in a more specialized journal in the present form. To deserve publication in Nature Communications a more extensive and quantitative agreement with the calculations should be shown.

Reply:

First, we thank the referee again for the valuable remarks. As commented by the referee, the novelty of our results is in the experimental RIXS spectra. After having included two additional measurements—RIXS data of an over-doped Pb-Bi2212 sample as a benchmark and detailed temperature-dependent measurements across T_c for OP Pb-Bi2212, our experimental results undoubtedly show an enhancement of the exciton intensity in the superconducting phase relative to the normal phase with a pseudogap. The observation is unprecedented and surprising. To the best of our knowledge, this atypical change in the spectral feature of excitons can only be explained by the new theoretical idea based on electron fractionalization. Although the TCFM is a simple model that describes purely the charge channel and does not take into account the spin excitation, we have reached a quantitative agreement between RIXS spectra and TCFM calculations; see the new discussions added to lines 376-416, pages 6-7 in section Discussion and Conclusion. We hope the referee agrees with the publication of the manuscript.

Reviewers' comments:

Reviewer #1 (Remarks to the Author):

The revised manuscript by Singh et al. has been considerably improved by the authors. In particular, the physical concept of electron fractionalization in the pseudogap state is now sufficiently made clear by adequately summarizing the previous theory papers by some of the coauthors in relation to the present experimental data. The description and explanation of the incident energy dependence of the experimental RIXS data has also significantly improved. The referee recognizes that additional RIXS data for several temperatures are presented that strengthen the experimental basis of the paper. However, the referee still disagrees with the authors that the agreement between the experimental data and the simulations is quantitative. The authors even implicitly admit that a difference occurs that possibly is due to extended tails of the dd excitations (and other high-energy excitations) and the paramagnon excitation. They suggest that subtracting these spectral contributions, not covered in their theory description, will clearly identify the exciton peak around 1 eV. Unfortunately, the authors do not present such a spectral deconvolution that could prove their assumption. The authors show, however, now an analysis of the spectral weight development, but miss to illustrate clearly how this spectral weight analysis was carried out in detail and how the error bars were determined which appear to be much too small compared to the noise level and statistical variation of the spectra.

In summary, the referee remains at the opinion that the paper reports interesting medium-energy scattering data by RIXS across the superconducting transition that are not rigorously analyzed and by this are not conclusively compared with the developed theory model. The paper is not of the quality and urgency required by Nature Communications, but appears now after the made improvements to be suitable for a more specialized journal (possibly from the Nature journal or partner journal series).

Reviewer #2 (Remarks to the Author):

In their revision and response to my previous report, the authors have gone to some length to improve their manuscript on various fronts. I especially appreciate their addition of the new variable-temperature measurements, as well as their analysis of the different spectral normalization schemes. I am now convinced that the spectral variation with temperature is a real effect, which has an empirical connection to the pseudogap/superconductivity physics, as demonstrated in the new Fig. 4. The fact that such a variation occurs at such a high energy scale is a somewhat surprising result that deserves to be known by the high- T_c community.

That being said, essentially all three referees noticed in the previous round that the most prominent spectral variation is in fact with the dd peak. While I previously pointed to the dd peak under the issue of spectral normalization, the other referees explicitly asked about its physical origin. I would therefore like to comment on the authors' response to Referee 1's pertinent remark, on page 7 of the rebuttal document. In my opinion, even when using an intensity normalization range that includes the dd peak (Fig. R1), it is still clear from the data that the *shape* of the dd peak changes between 50K and 120K. I believe that this will be clearly seen if the difference is taken. Moreover, the effect is best seen with photon energies slightly above the L3 edge, just like the excitons in the 1 eV region. In this sense, I tend to disagree with the authors' response to Referee 1, "The temperature dependence of the dd region may look as strong as that in the 1 eV region, but this is not the case, as evidenced by the new intensity normalization ..." I think the effect continues to look as strong as (and in fact, stronger than) the one in the 1 eV region.

I am thus unconvinced of the authors' approach of ignoring the dd peak's temperature dependence in their interpretation of the physics. In my opinion, a successful model should treat both the dd and the "1 eV region" on more or less equal footing, because they both presumably reflect a fundamental (and somewhat unexpected) change in the electronic structure. I do not think the measurement data as they now stand would allow the authors to unequivocally claim a strong support for the two-

component fermion model. As the authors remarked in response to Referee 1, the model at present cannot account for the upper Hubbard band.

To summarize, I consider this manuscript as containing interesting new observations. I believe that the work would be better appreciated if the experimental data can be presented with less association to the two-component fermion model, which does not offer a globally satisfactory description of the data after all. Whether a stand-alone experimental story would warrant publication in Nature Communications is a somewhat subjective question - I tend to say yes in the present case, because I find it surprising that the superconductivity/pseudogap physics has influence up to a rather high energy scale, which is generally under-appreciated in the community. I would not recommend the manuscript to be published in its present form because the interpretation is not sufficiently substantiated by the measurement data.

Reviewer #3 (Remarks to the Author):

The authors have submitted a new version of their manuscript that encompasses several relevant improvements, which were stimulated by the 3 referees. The reviewer had expressed their perplexity about the solidity of the link between the theoretical model calculations and the experimental results. The former had already been published before and is not to be re-discussed here. The latter constitute the original and significant content of this work. Although the situation has not been totally reversed, I am convinced that the manuscript has improved significantly.

I consider that the main weakness of the manuscript is still in pretending that the TCFM model is strongly supported by the RIXS results. However the abstract correctly states that, mainly, the results "impose a crucial constraint on theories for the pseudogap and superconducting mechanisms", which is a valid statement. The new experimental data at other temperatures and of an overdoped sample, presented in Figure 4, are indeed very relevant and strengthen significantly the phenomenology. Moreover, the responses to the referee #2 about the possible experimental artefacts are quite convincing in my opinion. Therefore, I can amend partly the concluding remarks of my previous reports. The experimental basis is now more complete and convincing, the phenomenology about the non-trivial T dependence of the high-energy exciton in RIXS spectra across the superconducting transition is relevant and deserves publication with good visibility. However, the link to the theory is in my opinion still questionable. I therefore suggest toning drastically down the claims of confirmation of the TCFM model. By presenting it as only one possible interpretation of the RIXS results, the theory would anyway hold the great merit of having stimulated an original experimental work. After such revision, I think that the manuscript might deserve publication in Nature Communications.

Reply to Referees

We greatly thank you for your valuable remarks, which led us to improve the quality of the manuscript significantly. In response to your comments, we have thoroughly revised the manuscript. The major revised parts are highlighted in blue in the main text. Our point-by-point replies to the referees' remarks are given in the following.

Reviewers' comments and our replies:

Comments of Reviewer #1:

The revised manuscript by Singh et al. has been considerably improved by the authors. In particular, the physical concept of electron fractionalization in the pseudogap state is now sufficiently made clear by adequately summarizing the previous theory papers by some of the coauthors in relation to the present experimental data. The description and explanation of the incident energy dependence of the experimental RIXS data has also significantly improved. The referee recognizes that additional RIXS data for several temperatures are presented that strengthen the experimental basis of the paper. **However, the referee still disagrees with the authors that the agreement between the experimental data and the simulations is quantitative.** The authors even implicitly admit that a difference occurs that possibly is due to extended tails of the dd excitations (and other high-energy excitations) and the paramagnon excitation. They suggest that subtracting these spectral contributions, not covered in their theory description, will clearly identify the exciton peak around 1 eV. **Unfortunately, the authors do not present such a spectral deconvolution that could prove their assumption.** The authors show, however, now an analysis of the spectral weight development, but **miss to illustrate clearly how this spectral weight analysis was carried out in detail and how the error bars were determined** which appear to be much too small compared to the noise level and statistical variation of the spectra.

In summary, the referee remains at the opinion that the paper reports interesting medium-energy scattering data by RIXS across the superconducting transition that are not rigorously analyzed and by this are not conclusively compared with the developed theory model. The paper is not of the quality and urgency required by Nature Communications, but appears now after the made improvements to be suitable for a more specialized journal (possibly from the Nature journal or partner journal series).

Reply:

We first thank the referee for his/her comments. We are grateful to the referee for appreciating our efforts in improving the manuscript. In the following, we rebut the referee's criticism that the agreement between the experimental data and the simulations is not quantitative. We also explain the data analysis on temperature-dependent spectral weights shown in Fig. 4c.

Referee's comment (1):

However, the referee still disagrees with the authors that the agreement between the experimental data and the simulations is quantitative. The authors even implicitly admit that a difference occurs that possibly is due to extended tails of the dd excitations (and other high-energy excitations) and the paramagnon excitation. They suggest that subtracting these spectral contributions, not covered in their theory description, will clearly identify the exciton peak around 1 eV. Unfortunately, the authors do not present such a spectral deconvolution that could prove their assumption.

Reply:

First, for the Cu L3-edge RIXS, the exciton feature ~ 1 eV is mixed with the strong tail of the dd excitation peak and the contributions from paramagnons and other excitations. One can still identify the exciton peak energy and compare the obtained exciton peak energy with the TCFM calculation. However, it is nontrivial to extract the spectral profile of the excitonic excitation. In contrast, the spectral background of exciton in the O K-edge RIXS does not contain paramagnons, the dd excitation is weaker and is, therefore, simpler. In the previous manuscript, we intended to say that one can extract the exciton peak of the O K-edge RIXS through a background subtraction and compare its spectral profile with the TCFM calculation as shown Figs. 5b and 5c. These comparisons led us to find a quantitative agreement between the experiment and calculation as explained in the manuscript in the following sense:

1. The resonance exciton energy is observed around the energy 1eV for the incident energy and the peak energy shifts proportionally with the incident energy region 0.8-1.5 eV.
2. The enhancement of the peak intensity is around 10% in the superconducting state as compared to the pseudogap state.

However, we admit that the comparison is not fully quantitative for the line shape. We emphasize that the enhancement of the peak in the present RIXS measurement is by itself a highly nontrivial result which deserves to be shared by wide readership. The TCFM analysis and the absence of enhancement in the conventional single-component systems offer, at least qualitatively, a possible interpretation of it, which also suggests a fundamental significance of the measured result to be shared by the people outside of the community as well. By considering this, we have revised in the last paragraph of the introduction (lines 107-125), the third paragraph of section Two-component fermion model analysis and the first, third and last paragraphs of section Discussion and conclusion (lines 386- 397 and lines 423-438, page 7 and lines 550-557, page 8) to respond to this comment.

Referee's comment (2):

The authors ... miss to illustrate clearly how this spectral weight analysis was carried out in detail and how the error bars were determined.

Reply:

The exciton spectral weight plotted in Fig. 4c is defined as the integration of the RIXS spectrum between 0.5 and 1.6 eV for OP and between 0.9 and 1.8 eV for OD samples, respectively. In this plot, the exciton spectral weight is normalized to one for 250 K and the

error bars are estimated from the variations in exciton spectral weight deduced by using different energy loss regions. Discussion on the estimate of the error bars is presented in Supplementary Information, responding to this comment. We have also revised the caption of Fig. 4c.

Comments of Reviewer #2:

In their revision and response to my previous report, the authors have gone to some length to improve their manuscript on various fronts. I especially appreciate their addition of the new variable-temperature measurements, as well as their analysis of the different spectral normalization schemes. I am now convinced that the spectral variation with temperature is a real effect, which has an empirical connection to the pseudogap/superconductivity physics, as demonstrated in the new Fig. 4. The fact that such a variation occurs at such a high energy scale is a somewhat surprising result that deserves to be known by the high-T_c community.

That being said, essentially all three referees noticed in the previous round that the most prominent spectral variation is in fact with the dd peak. While I previously pointed to the dd peak under the issue of spectral normalization, the other referees explicitly asked about its physical origin. I would therefore like to comment on the authors' response to Referee 1's pertinent remark, on page 7 of the rebuttal document. In my opinion, even when using an intensity normalization range that includes the dd peak (Fig. R1), it is still clear from the data that the *shape* of the dd peak changes between 50K and 120K. I believe that this will be clearly seen if the difference is taken. Moreover, the effect is best seen with photon energies slightly above the L3 edge, just like the excitons in the 1 eV region. In this sense, I tend to disagree with the authors' response to Referee 1, "The temperature dependence of the dd region may look as strong as that in the 1 eV region, but this is not the case, as evidenced by the new intensity normalization ..." I think the effect continues to look as strong as (and in fact, stronger than) the one in the 1 eV region.

I am thus unconvinced of the authors' approach of ignoring the dd peak's temperature dependence in their interpretation of the physics. In my opinion, a successful model should treat both the dd and the "1 eV region" on more or less equal footing, because they both presumably reflect a fundamental (and somewhat unexpected) change in the electronic structure. I do not think the measurement data as they now stand would allow the authors to unequivocally claim a strong support for the two-component fermion model. As the authors remarked in response to Referee 1, the model at present cannot account for the upper Hubbard band.

To summarize, I consider this manuscript as containing interesting new observations. I believe that the work would be better appreciated if the experimental data can be presented with less association to the two-component fermion model, which does not offer a globally satisfactory description of the data after all. Whether a stand-alone experimental story would warrant publication in Nature Communications is a somewhat subjective question - I tend to say yes in the present case, because I find it surprising that the superconductivity/pseudogap physics has influence up to a rather high energy scale, which is generally under-appreciated in the community. I would not recommend the manuscript to be published in its present form because the interpretation is not sufficiently substantiated by the measurement data.

Reply:

We first thank the referee for his/her comments. We are pleased the referee is convinced that the spectral variation with temperature is a real effect. In the following, we reply to the referee's comments on our previous reply to Referee 1 and the suggestion to publish the paper as a stand-alone experimental story.

Referee's comment (1):

I tend to disagree with the authors' response to Referee 1, "The temperature dependence of the *dd* region may look as strong as that in the 1 eV region, but this is not the case, as evidenced by the new intensity normalization ..." I think the effect continues to look as strong as (and in fact, stronger than) the one in the 1 eV region. I am thus unconvinced of the authors' approach of ignoring the *dd* peak's temperature dependence in their interpretation of the physics. In my opinion, a successful model should treat both the *dd* and the "1 eV region" on more or less equal footing.

Reply:

By mistake, we wrote that sentence. We agree that the enhancement in the energy range of the *dd* excitations is as large as the exciton ~ 1 eV. Nevertheless, the conclusion and other discussions of that reply were correct. The manuscript indeed addresses this through discussing the "shake-up" effect. In addition, the physics involved with the *dd* peak's temperature dependence is included now in the revised manuscript. Since the hole resides in the same coherent LHB, the physics of electron fractionalization will equally apply to an exciton with the electron in the upper Hubbard band. Surely one would like to have a model which offers a globally satisfactory description of the data. The extension of the TCFM to include the upper Hubbard band, however, requires a substantial extension of the model. The much stronger effect of the core-hole potential on the UHB than on the IGB in the RIXS intermediate state needs to be included, and a "hybridization" term between the UHB and IGB has to be introduced. Parameters needed for such an extension are not available so far from the ARPES data, because the data are limited to the energy region closer to the Fermi level. Instead, we have included the above qualitative discussion to the revised manuscript. See the discussions in the 5th paragraph of section Discussion and conclusion (line 482 - line 491, pages 7 and 8).

Referee's comment (2):

the work would be better appreciated if the experimental data can be presented with less association to the two-component fermion model, which does not offer a globally satisfactory description of the data after all.

Reply:

Again, we thank the referee for agreeing that the unusual enhancement of exciton is real and surprising. In principle, one could present the data as a stand-alone experimental paper. In such an approach, however, physical consequences and significance of the result become less clear in the sense how the result challenges conventional understanding. To the

best of our knowledge, no explanations other than the TCFM are available so far. We, however, agree that the interpretation of the experimental result by the TCFM may not be unique and the present interpretation by the TCFM is neither complete nor fully quantitative. To meet the referee's comment we have revised the manuscript thoroughly and carefully to address that the TCFM offers just one possibility of understanding for the experimental result and it is open to other interpretations as well if possible. We have revised the manuscript to present the experimental data as well as discussions involved with electron fractionalization along this line. To clearly points out this, we have revised, including: the last sentence of the abstract, the last paragraph of the introduction (lines 107-125), the first paragraph of section Two-component fermion model analysis (lines 310-328), the third and last paragraphs of section Discussion and conclusion (lines 423-438 and lines 550-557). We hope the referee would agree with our approach.

Comments of Reviewer #3:

The authors have submitted a new version of their manuscript that encompasses several relevant improvements, which were stimulated by the 3 referees. The reviewer had expressed their perplexity about the solidity of the link between the theoretical model calculations and the experimental results. The former had already been published before and is not to be re-discussed here. The latter constitute the original and significant content of this work. Although the situation has not been totally reversed, I am convinced that the manuscript has improved significantly.

I consider that the main weakness of the manuscript is still in pretending that the TCFM model is strongly supported by the RIXS results. However the abstract correctly states that, mainly, the results “impose a crucial constraint on theories for the pseudogap and superconducting mechanisms”, which is a valid statement. The new experimental data at other temperatures and of an overdoped sample, presented in Figure 4, are indeed very relevant and strengthen significantly the phenomenology. Moreover, the responses to the referee #2 about the possible experimental artefacts are quite convincing in my opinion. Therefore, I can amend partly the concluding remarks of my previous reports. The experimental basis is now more complete and convincing, the phenomenology about the non-trivial T dependence of the high-energy exciton in RIXS spectra across the superconducting transition is relevant and deserves publication with good visibility. However, the link to the theory is in my opinion still questionable. I therefore suggest toning drastically down the claims of confirmation of the TCFM model. By presenting it as only one possible interpretation of the RIXS results, the theory would anyway hold the great merit of having stimulated an original experimental work. After such revision, I think that the manuscript might deserve publication in Nature Communications.

Reply:

We first thank the referee for his/her valuable comments. We appreciate that the referee agrees that the manuscript has been improved significantly. In response to the referee's comments, we have revised the manuscript carefully to emphasize that the TCFM, which motivated the present work, offers just one possible interpretation. The present experimental result also calls for other possible interpretations if possible. To deepen the

quantitative analysis on the TCFM side will be our future work because the comparison between the experimental result and the TCFM interpretation is not quantitative enough yet. Now, the title of the revised manuscript has been changed and the concluding sentence of the abstract clearly points out that the TCFM approach provides a possible interpretation. We have also examined the main text carefully and thoroughly and revised it along this line, including the last paragraph of the introduction (lines 107-125), the first paragraph of section Two-component fermion model analysis (lines 310-328), as well as the third and last paragraphs of Discussion and conclusion section (lines 423-438 and lines 550-557).

Reviewers' comments:

Reviewer #1 (Remarks to the Author):

The manuscript by Singh et al. has been revised a second time to reduce the authors claims of agreement of the experimental RIXS data with the two-component fermion model. The referee agrees that the authors attempt to provide a more honest assessment of the current understanding of the experimental data and their relation to one possible theory explanation. However, the referee still finds it inappropriate that the authors still insist in the main text of the paper on a quantitative agreement between experiment and calculation (see line 397).

Reviewer #2 (Remarks to the Author):

I have re-reviewed the manuscript. I think the authors did a good job in revising their manuscript, quite thoroughly indeed, such that the experimental data stand out with less mandatory association with the two-component Fermion model. The limitation of the model in quantitatively describing the experimental data is now discussed more openly. Meanwhile, I tend to agree with the authors that the model is probably the best available one despite its limitations, and I continue to think that a pronounced spectral change at energies far above the superconducting gap is a significant result that will trigger broad interest in the high- T_c community. I recommend the work for publication in Nature Communications.

Reviewer #3 (Remarks to the Author):

The new edition of the manuscript complies with my previous requests. I support its publication in Nature Communications.

Response to the Referees

(Original reviewers comments are in black italic fonts; our responses are in blue.)

Reviewer #1 (Remarks to the Author):

The manuscript by Singh et al. has been revised a second time to reduce the authors claims of agreement of the experimental RIXS data with the two-component fermion model. The referee agrees that the authors attempt to provide a more honest assessment of the current understanding of the experimental data and their relation to one possible theory explanation. However, the referee still finds it inappropriate that the authors still insist in the main text of the paper on a quantitative agreement between experiment and calculation (see line 397).

Our response:

We thank the referee for the remark. In response to this comment, we have removed the wording “quantitative” in the first paragraph of the Discussion section. Now our revised manuscript presents a consistent tone that, although the TCFM explains the enhancement of the excitonic excitation, we have not yet reached a fully quantitative agreement between measurements and theoretical calculations.

Reviewer #2 (Remarks to the Author):

I have re-reviewed the manuscript. I think the authors did a good job in revising their manuscript, quite thoroughly indeed, such that the experimental data stand out with less mandatory association with the two-component Fermion model. The limitation of the model in quantitatively describing the experimental data is now discussed more openly. Meanwhile, I tend to agree with the authors that the model is probably the best available one despite its limitations, and I continue to think that a pronounced spectral change at energies far above the superconducting gap is a significant result that will trigger broad interest in the high- T_c community. I recommend the work for publication in Nature Communications.

Our response:

We thank the referee for his/her comments.

Reviewer #3 (Remarks to the Author):

The new edition of the manuscript complies with my previous requests. I support its publication in Nature Communications.

Our response:

We thank the referee for his/her support.